

# Nonlocal correlations in noisy multiqubit systems simulated using matrix product operators

## Haggai Landa[1][*] and Grégoire Misguich[2][†]

**1** IBM Quantum, IBM Research – Israel, Haifa University Campus,
Mount Carmel, Haifa 31905, Israel
**2** Institut de Physique Théorique, Université Paris-Saclay,
CEA, CNRS, 91191 Gif-sur-Yvette, France

[*] haggai.landa@ibm.com , [†] gregoire.misguich@ipht.fr

## Abstract

We introduce an open-source solver for the Lindblad master equation, based on matrix product states and matrix product operators. Using this solver we study the dynamics of tens of interacting qubits with different connectivities, focusing on a problem where an edge qubit is being continuously driven on resonance, which is a fundamental operation in quantum devices. Because of the driving, induced excitations propagate through the qubits until the system reaches a steady state due to the incoherent terms. We find that with alternating-frequency qubits whose interactions with their off-resonant neighbors appear weak, the tunneling excitations lead to large correlations between distant qubits in the system. Some two-qubit correlation functions are found to increase as a function of distance in the system (in contrast to the typical decay with distance), peaking on the two edge qubits farthest apart from each other.

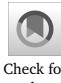

# 1   Introduction

Understanding the dynamics of interacting quantum systems subject to dissipative processes has become a corner stone in many fields of research [1–21]. Such dynamical systems can only rarely be solved analytically, and numerical methods are an essential tool for detailed studies and a better understanding. While the solution for a few coupled quantum systems with low-dimensional Hilbert spaces can be obtained using brute-force integration, scaling up numerical studies to larger systems is in general exponentially expensive in the computer memory required for the storage of quantum states.

In order to cope with the exponential increase in the computing resources when solving quantum problems exactly, some approaches take advantage of the compression offered under certain conditions by a representation of states using tensor networks (TN) [22, 23]. The simplest TN states are matrix-product states (MPS). This representation is at the core of the density-matrix renormalization group (DMRG) algorithm [24, 25], which proved to be a powerful method to obtain ground (and also excited) states of Hamiltonians with short-ranged interactions in one dimension (1D), as well as in some two-dimensional (2D) systems [26]. MPS have not only been used to study static properties, they also allowed developing algorithms for simulating Hamiltonian dynamics [27–29].

Applying the above matrix-product ideas to operators naturally leads to matrix-product operators (MPO) and matrix product density operators (MPDO) [30]. TN representations can be used to encode the many-body density matrix describing a mixed state and to study the dynamics of the Lindblad master equation, describing a system coupled to a Markovian bath [30–37]. A majority of the studies considered one-dimensional (1D) problems, for which matrix-product

representations are particularly adapted, but some results could also be obtained for in other geometries [34, 35, 37, 38].

In particular, the study of nonequilibrium dynamics in large inhomogeneous systems with no symmetries to exploit is relevant for the development of devices for quantum computation and quantum technologies [39]. Various quantum processors and simulators already reach tens and in some cases hundreds of qubits [40]. Efforts in the field are focused both on applications with noisy intermediate-scale systems, and on fault tolerant quantum computing envisioned in the longer run [41, 42]. In both cases, the interplay of noise and quantum correlations in the dynamics of the many-qubit state is an issue of crucial importance with a lot of open questions.

In this work we introduce an MPO-based high-performance numerical solver for density matrix dynamics obeying the Lindblad master equation. The presented open-source package [43], called lindbladmpo, allows users to simulate the dynamics of (two-level) qubits, in a uniform rotating frame (i.e., with time-independent coefficients in the master equation). General single-qubit Hamiltonian parameters are supported, together with two-qubit interactions [of "flip-flop" (XY) and Ising (ZZ) coupling form] with arbitrary connectivity, and three dissipative jump operators that describe energy exchange with a thermal bath and dephasing. The solver employs internally the open-source library ITensor [44], and simulations with tens to hundreds of qubits in 1D and 2D setups using an earlier prototype of this solver are reported in [34, 35].

Using this solver, we study qubits with XY interactions where an edge qubit is being continuously driven on resonance. Such a resonant driving of single qubits is the fundamental workhorse of quantum information processing in many qubit devices, being employed to realize single-qubit rotations and also to generate entanglement in some setups. The driven dynamics are often studied by focusing on very small qubit systems, however, the focus of the current work is on the many body nonlocal dynamics induced by driving a single qubit. Indeed, due to the driving, excitations are induced on top of the system's ground state and those excitations (that can be seen as quasiparticles) propagate in the lattice, strongly influenced by the qubit frequencies, their connectivity, and dissipation parameters. We study the dynamics of the spreading excitations up to the steady state that the energy relaxation induces.

We focus on two main setups – identical qubits in a linear chain, and qubits with alternating frequencies (between nearest neighbors), in a plaquette configuration which is essentially a ring with two additional edge qubits. The one-dimensional chain setup forms a natural starting point and is relevant to identical qubits, and we study it mostly as a "warm-up" problem that allows us to portray a picture of the driven dynamics in the model and to understand the limitations of the solver. The setup with alternating-frequency qubits on a plaquette is motivated by currently deployed IBM Quantum devices [45] accessible via the cloud using Qiskit [46], whose time-dependent Hamiltonian can be programmed [47]. In those devices the qubit connectivity is that of a "heavy-hexagonal" lattice [48, 49], which consists of connected qubit plaquettes. Typically, the fixed-frequency qubits are purposefully chosen to have different frequencies for neighboring ones in order to reduce unwanted interactions, and the design of qubit frequency patterns and their effect on the dynamics is an active field of research [50–52].

Emphasizing that our study does not constitute a simulation of such devices but rather a first step in that direction, the results that we find unravel interesting effects. Even with off-resonant qubits that appear to be only weakly interacting, rich nonlocal entanglement dynamics between the driven qubit and distant qubits can be seen. Although the entanglement in the system dies after a typical time (related to the extension of the system and the dissipative processes), large two-qubit correlations can form between distant qubits. Moreover, we find cases in which some correlation functions increase with distance in the system, peaking on

the two qubits farthest apart from each other. Varying the number of qubits, the magnitude of the maximal correlation initially increases as a function of the system size, and eventually decreases.

We have studied also the robustness of the observed correlations in the presence of dephasing, variations in the drive strength, and additional ZZ qubit coupling. Uncontrolled nonlocal correlations of qubits in large devices could be detrimental to their usage for computational tasks and are therefore important to simulate and explore as a function of the parameters. In addition to the specific results highlighting the formation of nonlocal correlations in the setup that we investigate, this work serves to demonstrate the usage of the solver in studying a dynamical problem with large systems and extensive parameter dependence. We discuss aspects important when undertaking such a systematic analysis, which is performed here in conjunction with the Python package `qiskit-dynamics` [53,54] – employed for simulations with up to ten qubits. The source code used to generate the simulations and analysis presented in this work is included as an example in the solver repository [43].

In Sec. 2 the model supported by the solver and its initial state and observables are specified. In Sec. 3 and Sec. 4 we present the results from the detailed numerical studies described above. In Sec. 5 we summarize the results and discuss our conclusions and potential outlook for further research. In the Appendix we present more details of the numerical study and the numerical verification of the results presented in this paper.

## 2 Model Setup

### 2.1 Lindbladian

We model $N$ interacting qubits with an arbitrary connectivity. The qubits are two-level systems, and we use the three Pauli matrices and ladder operators with the sites indexed by $i$,

$$\sigma_i^a, \qquad a = \{x, y, z\}, \qquad \sigma_i^{\pm} = (\sigma_i^x \pm i\sigma_i^y)/2. \tag{1}$$

The state of an open quantum system is defined by a density matrix $\rho$. We consider time evolution described using the master equation

$$\frac{\partial}{\partial t}\rho = \mathcal{L}[\rho] \equiv -\frac{i}{\hbar}[H, \rho] + \mathcal{D}[\rho], \tag{2}$$

where $[\cdot, \cdot]$ is the commutator of two operators, and the Liouvillian (or Lindbladian) $\mathcal{L}$ is a (linear) superoperator acting on the operator $\rho$. The unitary evolution due to interactions and coherent driving terms is generated by the Hamiltonian $H$, while $\mathcal{D}[\rho]$ is a superoperator (sometimes known as the dissipator) that accounts for incoherent dephasing and relaxation processes due to the environment.

The Hamiltonian is defined in the rotating frame with respect to a uniform frequency and all parameters defined below are assumed constant in this frame. The details of the lab frame and the transformation are given in App. G. Decomposing the Hamiltonian in the rotating frame into the sum of on-site terms and the interaction part $K$, we have

$$H/\hbar = \sum_i \frac{1}{2}\left[h_{z,i}\sigma_i^z + h_{x,i}\sigma_i^x + h_{y,i}\sigma_i^y\right] + K, \tag{3}$$

with the interaction being a sum of two-qubit terms,

$$K = \sum_{i \neq j}\left(J_{ij}\sigma_i^+\sigma_j^- + \text{H.c.} + \frac{1}{2}J_{ij}^z\sigma_i^z\sigma_j^z\right) = \frac{1}{2}\sum_{i \neq j}\left(J_{ij}\sigma_i^x\sigma_j^x + J_{ij}\sigma_i^y\sigma_j^y + J_{ij}^z\sigma_i^z\sigma_j^z\right). \tag{4}$$

We treat three typical dissipator terms in Eq. (2),

$$\mathcal{D} = \sum_{j=0}^{2} \mathcal{D}_j, \tag{5}$$

with

$$\mathcal{D}_0[\rho] = \sum_i g_{0,i} \left( \sigma_i^+ \rho \sigma_i^- - \frac{1}{2} \{ \sigma_i^- \sigma_i^+, \rho \} \right), \tag{6}$$

$$\mathcal{D}_1[\rho] = \sum_i g_{1,i} \left( \sigma_i^- \rho \sigma_i^+ - \frac{1}{2} \{ \sigma_i^+ \sigma_i^-, \rho \} \right), \tag{7}$$

$$\mathcal{D}_2[\rho] = \sum_i g_{2,i} \left( \sigma_i^z \rho \sigma_i^z - \rho \right). \tag{8}$$

The physical process corresponding to Eqs. (6)-(7) are incoherent transitions from 0 to 1 and vice versa, which could be caused by energy exchange with a thermal bath. Eq. (8) corresponds to dephasing in $xy$ plane.

It should be noted that in a one-dimensional geometry where a Jordan-Wigner transform can be used, the Lindblad dynamics of the above spin models can be solved using the covariance matrix methods in a few cases (the so called quasi-free and quadratic Lindblad master equations). This is the case in presence of XY couplings $J_{ij}$ between nearest-neighbor qubits only, and arbitrary magnetic field $h_{z,i}$ (but with $J_{ij}^z = 0$ and $h_{y,i} = h_{x,i} = 0$). For a recent work on this topic see Ref. [55].

## 2.2 Initial state

The simulator takes as input an initial state (at $t_0$) that can be a pure Pauli product state,

$$\rho(t_0) = |\psi_0\rangle\langle\psi_0|, \qquad |\psi_0\rangle = \prod_i |\pm a_i\rangle, \tag{9}$$

with $|\pm a_i\rangle$ and $a_i \in \{x, y, z\}$ a Pauli eigenstate of qubit $i$;

$$\sigma_i^a |\pm a_i\rangle = \pm |\pm a_i\rangle. \tag{10}$$

Another possibility is a graph state defined by specifying a set $V$ of pairs of qubits, to all of which a controlled-$Z$ gate is applied, after initializing all qubits along $x$,

$$|\psi_0\rangle = \prod_{(j,k)\in V} CZ[j,k] \prod_i |+x_i\rangle. \tag{11}$$

In addition, the simulator can also save its internal state representation to a file, and load a saved state for use as the initial state of a subsequent simulation.

## 2.3 Observables

For the initial time $t_0$, final time $t_f$, and intermediate times $t_k = t_0 + m\tau, m \in \mathbb{N}$ (defined using the solution's fixed time step $\tau$), the output of the simulator consists of

1. Single-qubit (1Q) observables,

$$\langle \sigma_i^a(t_k) \rangle. \tag{12}$$

2. Two-qubit (2Q) observables,

$$\left\langle \sigma_i^a(t_k) \sigma_j^b(t_k) \right\rangle. \tag{13}$$

These are also referred to as Pauli expectation values (of one and two qubits, respectively), at time $t_k$.

## 2.4 Global characteristics

In addition to the 1Q-2Q observables, one can define a few global characteristics of the state;

1. The trace of the density matrix, which should be conserved (and remain equal to 1),

$$\text{tr}\{\rho\}. \tag{14}$$

2. The second Rényi entropy, which is closely related to the purity of the density matrix,

$$S_2 \equiv -\ln\text{tr}\{\rho^2\}. \tag{15}$$

3. The operator space entanglement entropy (OSEE) [56] for a bipartition at the central bond. The OSSE associated to a given bipartition is defined by considering the vectorization operation. The vectorized state that is denoted as

$$|\rho\rangle\rangle, \tag{16}$$

is a pure state living in an enlarged Hilbert space where the local space is spanned by the 3 Pauli matrices plus the identity matrix, as defined in Eq. (42). The OSEE associated to a given bipartition into two subsystems $A$ and $B$ is by definition the von Neumann entanglement $S_{\text{vN},|\rho\rangle\rangle}^{(A)} = S_{\text{vN},|\rho\rangle\rangle}^{(B)}$ associated to this partition of the vectorized pure state $|\rho\rangle\rangle$,

$$S_{\text{OSEE}} = -\text{tr}\left\{\rho^A \ln\rho^A\right\}, \tag{17}$$

where

$$\rho^A \equiv \text{tr}_B\{|\rho\rangle\rangle\langle\langle\rho|\}. \tag{18}$$

This quantity vanishes for (mixed or pure) product states. In the case of a pure state the OSEE of a subsystem is twice the Von Neumann entropy of that subsystem.

## 2.5 Solver scope

By construction an MPO (or MPS) representation associates some matrix to each qubit, and the state is obtained by multiplying these matrices (see Eqs. 41 and 42). This means that one has to choose a numbering of the qubits, and this numbering defines in which order the matrices should be multiplied. On the other hand, the qubits have some specific connectivity in the device, and the numbering above means that one chooses a path that goes through every qubit. This effectively maps the problem onto a 1D problem with (potentially) long-ranged interactions along the MPO path. This approach is particularly efficient when the geometry of the couplings is 1D, but, importantly, it can apply to other geometries. With this MPO representation the computational resources needed for a simulation are a function of the amount of correlations, as well as on the geometry of the couplings in the system. The number of qubits feasible for the MPO solver therefore depends on the Hamiltonian parameters, the dissipation, the topology (qubit connectivity) and the time and precision required. As an illustration, consider a 2D geometry with $N = L_x \times L_y$ qubits, and a snake-like MPO path elongated in the $x$ direction and zigzaging in the transverse $y$ direction. For an illustration of such MPO path, see for instance Fig. 25 in [35]. For states where the OSEE – which measures the amount of bipartite correlation in the system (see Sec. 2.4) – obeys an *area law*,[1] the

---

[1]Area law means that the OSEE of a large subsystem scales like the size of its boundary. This is a mixed state counterpart of the area law of the entanglement entropy in pure states [57].

computational resources grow linearly in $L_x$ and exponentially in $L_y$. This can provide an important advantage over a brute-force approach that would scale exponentially in $N$.

The solver is written in C++ and can take advantage of multithreaded implementations of the LAPACK linear algebra library. It exposes a command-line interface as well as an easy-to-use Python wrapper with rich plotting capabilities. The solver uses the open source library ITensor [44] for the storage, time evolution and manipulation of the many-qubit state using MPS and MPO. The solver integrates the dynamics in fixed time steps using a Trotter expansion of order up to the fourth [58], adapted from the method $W^{\mathrm{II}}$ of [59], that allows for a compact MPO representation of the evolution operator in the presence of nonlocal interactions (along the one-dimensional path used to encode the density matrix).

In the next two sections we use the MPO solver to study a specific problem and consider the effect of different qubit connectivities and model parameters. The convergence of the results as a function of the model parameters and as a function of the numerical solver parameters is also analyzed.

# 3 Driving an edge qubit in an XY model

## 3.1 Problem Setup

We consider $N$ qubits coupled with their nearest neighbors according to a connectivity map that is left arbitrary at this point. The qubit frequencies $\omega_i$ in the lab frame alternate between two values, such that coupled qubits (nearest neighbors) always differ in frequency by

$$\nu_{ij} = \omega_i - \omega_j = \pm\Delta\,, \qquad \Delta = \omega_1 - \omega_0\,. \tag{19}$$

We choose this setup as an example of inhomogeneous system that can be parametrized using a single parameter, $\Delta$, which is the frequency difference between the qubits. It can be positive, negative, or zero if the qubits are all identical. Note however that the solver can in principle treat more general situations, with arbitrary $\omega_i$.

In the lab frame only qubit 0 is driven harmonically with an amplitude parametrized by $\Omega$, at a frequency resonant with its energy level spacing. Setting $\hbar = 1$ throughout the rest of the paper, the Hamiltonian in the lab frame is

$$H_{\mathrm{lab}} = \Omega\sigma_0^x \cos(\omega_0 t) + \frac{1}{2}\sum_i \omega_i(1 - \sigma_i^z) + K, \tag{20}$$

where $\omega_i > 0$, and putting a negative sign in front of $\sigma_i^z$ is a common convention for superconducting qubits. The interaction part is

$$K = J\sum_{\langle i,j\rangle}\left(\sigma_i^+\sigma_j^- + \mathrm{H.c.}\right) = \frac{1}{2}J\sum_{\langle i,j\rangle}\left(\sigma_i^x\sigma_j^x + \sigma_i^y\sigma_j^y\right), \tag{21}$$

corresponding to nearest-neighbor flip-flop (XY) coupling of a uniform magnitude. We fix the coupling coefficient to

$$J = 2\pi \times 1\,, \tag{22}$$

which sets the units of frequency and time.

Transforming to the frame rotating uniformly with the frequency of the driven qubit, defined by the unitary

$$R_0 = \exp\left\{-it\frac{1}{2}\omega_0\sum_i \sigma_i^z\right\}, \tag{23}$$

we obtain using the expansion in App. G (in the rotating-wave approximation), the time-independent Hamiltonian of Eq. (3), where the parameters are

$$h_{x,i} = \Omega \delta_{i,0}, \qquad h_{z,i} = (\omega_i - \omega_0) \in \{0, \Delta\}, \tag{24}$$

with all other parameters being 0. With the transformation of Eq. (23), the interaction Hamiltonian $K$ of Eq. (21) carries over to the rotating frame without change, and so do the dissipators of Eqs. (6)-(8), whose parameters we take to be uniform,

$$g_{0,i} = g_0, \qquad g_{1,i} = 0, \qquad g_{2,i} = g_2. \tag{25}$$

We note that the parameters $g_{0,i}$ describe transfer of population to the eigenstate of $\sigma_i^z$ with eigenvalue $+1$, which is in the product ground state of the Hamiltonian in Eq. (20) in the absence of driving. In qubit devices the environment temperature is typically significantly lower than the qubit's energy gap, and hence keeping the spontaneous emission terms only and neglecting the energy excitation process is typically an accurate approximation [60].

## 3.2 Identical qubits in a one-dimensional chain, with unitary dynamics

In preparation for the study of long qubit chains, in this subsection we examine the dynamics of a 1D chain with open boundary conditions, as depicted with $N = 9$ qubits in Fig. 1. As a warmup we consider a coherent/unitary evolution (no dephasing, no dissipation), with qubits that have identical frequencies in the lab frame ($\omega_i = \omega_0$), and we fix the driving amplitude of qubit 0 (at the left edge of the chain), such that the parameters of Eqs. (24)-(25) take the values

$$\Omega = 2\pi \times 1, \qquad \Delta = 0, \tag{26}$$

and

$$g_0 = g_2 = 0. \tag{27}$$

We employ essentially exact, brute-force simulations based on storing the full $2^N \times 2^N$ density matrix in memory. We evolve the dynamics with standard algorithms (e.g., Runge-Kutta) using the Python packages `qiskit-dynamics` [53] and `scipy` [61].

Figure 2 shows some curves of the Pauli $Z$ expectation value of a few qubits as a function of the time (left), and also a space-time diagram of the dynamics (right). Starting from an initial state where $\langle Z \rangle = 1$ for all qubits (coded in yellow in the right panel of Fig. 2), the drive on qubit 0 reduces its $z$-alignment and it attains a $Z$ mean value of $\sim 0.5$ up to $t \sim 1.5$. The interaction part $K$ of the Hamiltonian [Eq. (21)] conserves the total $z$ magnetization ($\sum_i \sigma_i^z$), but it allows the swapping of two neighboring qubit states. As a result, $\langle Z \rangle$ of the neighbors of qubit 0 start to decrease too. This propagation is ballistic and gives rise to a light-cone pattern (light green region adjacent to the yellow one in the right panel of Fig. 2), where each qubit

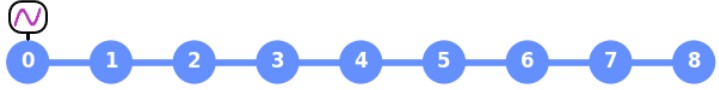

Figure 1: A linear chain of nine qubits with open boundary conditions. The uniform color of the qubits indicate that their lab frequencies are all identical [$\Delta = 0$ in Eq. (19)]. Qubit 0 on the left edge is being driven periodically as in Eq. (20), and the qubits interact with with their nearest neighbors by a "flip-flop" (XY) interaction term as in Eq. (21).

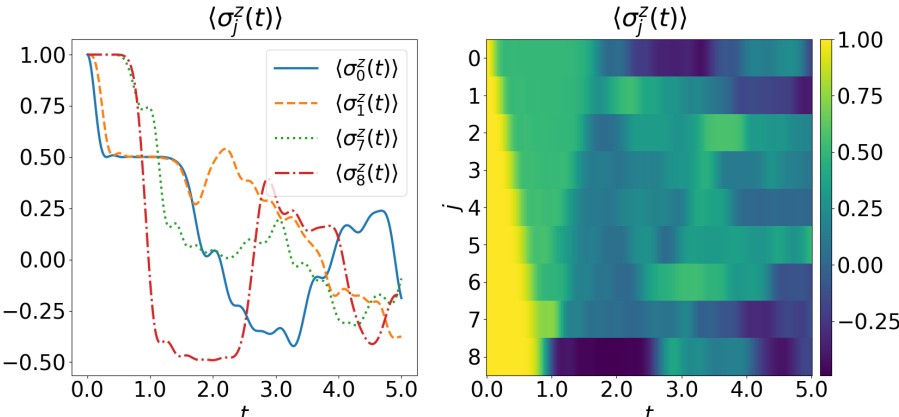

Figure 2: The Pauli $Z$ expectation values vs. time for nine qubits in a 1D chain as in Fig. 1, shown for the two qubits at each edge of the chain (left panel), and using a space-time diagram for all qubits (right panel). Qubit 0 is being driven according to Eq. (20), with the dynamics solved and presented in the rotating frame of Eq. (23). The parameters of Eqs. (24)-(25) are given in Eqs. (26)-(27). The color bar on the right associates the scale of colors to the expectation value of each qubit, plotted in rows from $j = 0$ to $j = 8$, for the time varying from $t = 0$ to $t = 5$. The propagation of the information signal from the driven qubit manifests as a light-cone in $\langle Z \rangle$ – see the text for a detailed discussion.

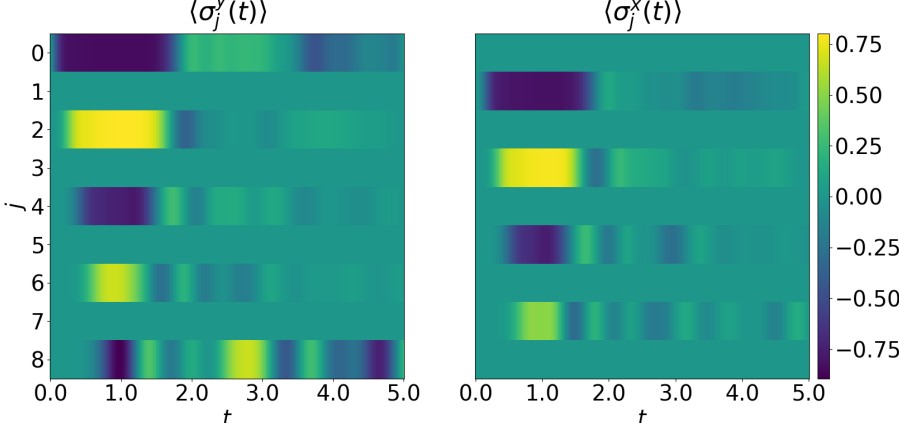

Figure 3: A space-time diagram of the $Y$ (left) and $X$ (right) Pauli observables of nine qubits in a 1D chain as in Fig. 1, in the same simulation as in Fig. 2. It can be seen that with qubit 0 being driven about its $x$ axis, its $X$ expectation value remains identically 0, while in $\langle Y \rangle$ it quickly settles to a large negative value, nearly constant for a relatively long time – until the signal reflecting back from the chain right edge returns. The angle of the single-qubit projection in $xy$ plane changes by $-\pi/2$ between neighboring qubits along the chain.

remains in the initial ground state up to a time linear in its distance from the driven qubit, and only then its $Z$ expectation value starts decreasing.

This behavior can be understood by realizing that, using the Jordan-Wigner transformation, the XY interaction in 1D maps onto a free-fermion Hamiltonian. In a linear chain with open boundary conditions one can indeed rewrite Eq. (21) as

$$K = J \sum_i \left( \sigma_i^+ \sigma_{i+1}^- + \text{H.c.} \right) = J \sum_i \left( c_i^\dagger c_{i+1} + \text{H.c.} \right), \tag{28}$$

where $c_i^\dagger$ and $c_i$ are creation and annihilation operators of a spinless fermion at lattice site $i$. The above Hamiltonian can be diagonalized in Fourier space, leading to

$$K = \sum_k \epsilon(k) c_k^\dagger c_k, \qquad \epsilon(k) = 2J\cos(k), \tag{29}$$

where $c_k^\dagger$ is the Fourier transform of $c_i^\dagger$ and the cosine gives the dispersion relation. From this it is standard to deduce the group velocity

$$v(k) = \frac{d}{dk}\epsilon(k) = -2J\sin(k), \tag{30}$$

of the (fermionic) excitations, or quasiparticles. This velocity is maximum around $k = -\pi/2$, giving a "speed of light" $v_{\max} = 2J$. The boundary of the light cone shown in Fig. 2 is indeed consistent with $v_{\max} \simeq 4\pi$, following the value of $J$ in Eq. (22).

Moreover, one can identify in the light-cone region a simple pattern for the mean direction of each qubit. In Fig. 3 it can be seen that qubit 0 becomes highly aligned with the negative $y$ axis, (remembering that it is being rotated about the $x$ axis in the lab frame), and the chain develops an alternating pattern of neighboring qubits, each rotated by $-\pi/2$ with respect to its neighbor on the left. The end qubit (number 8, at the right edge) rotates and attains its minimal $Z$ projection of $\sim -0.5$ starting at $t \sim 1$, remaining nearly constant up to $t \sim 2.5$ (accompanied by some $\langle Y \rangle$ oscillations). This long dip in $\langle Z \rangle$ can be seen to end after a reflection of the front of the information signal form qubit 8 has travelled back (left) in the chain to qubit 0 and reflected again reaching qubit 8 at $t \sim 2.5$. A similar phenomenon can be observed in the unitary evolution of a spin chain with the Hamiltonian of Eq. (21) when the initial state has some spatially inhomogeneous $\langle Z \rangle$. As shown in [62], in such a case the evolution of the magnetization is well described by a hydrodynamical picture where waves in the $\langle Z \rangle$ profile travel through the system and bounce at the edges of the chain. In the next subsection we study the configurations resulting from a "freezing" of the hydrodynamic picture when energy relaxation is introduced.

### 3.3   Identical qubits in a one-dimensional chain, with energy relaxation

We now turn to study the effect of energy relaxation on the model studied above, in chains with up to 31 qubits. We leave unchanged the coherent (Hamiltonian) parameters of Eq. (24) with the values used above in Sec. 3.2

$$\Omega = 2\pi \times 1, \qquad \Delta = 0. \tag{31}$$

We start by setting the incoherent parameters of Eq. (25) for the simulations presented below, to

$$g_0 = 0.1, \qquad g_2 = 0. \tag{32}$$

This implies a slow energy relaxation (on the scale of the dynamics), and no dephasing. The first step that we take involves benchmarking the numerical parameters that are optimal for MPO simulations of the given problem. This comprises an initial study of the dynamics, comparing MPO simulations to essentially exact brute-force numerics for low enough qubit numbers, and employing various checks of the convergence of observables for qubit numbers not accessible to exact simulations. This process is described in App. B, and we use the numerical parameters derived there (verified for up to $N = 61$ qubits) for obtaining the results presented in this subsection.

In Fig. 4 the $Z$ expectation value of the end qubit (the one in the right edge, furthest from the driven one), is plotted for several $N$ values up to $t = 5$. The near self-similarity of

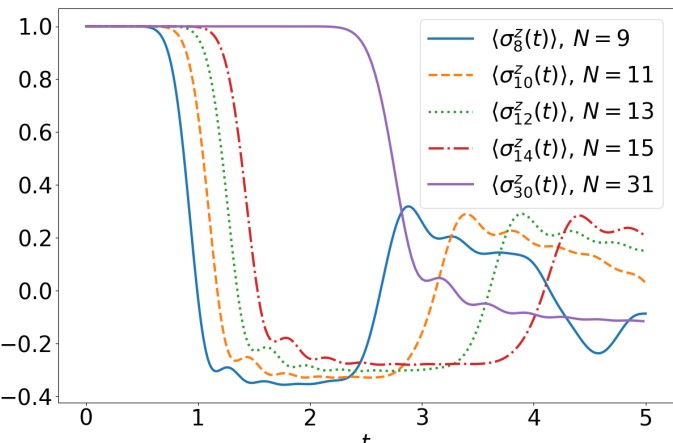

Figure 4: The $Z$ expectation value vs. $t$, of the qubit at the right end of the chain (at the other edge from the driven qubit), for chains of varying length from $N = 9$ to $N = 31$. The parameters are given in Eq. (31) and Eq. (32). The near self-similarity with shifts in time and $Z$ depth (linearly dependent on $N$) can be seen. See the text for a detailed discussion.

the curves for increasing $N$ is clear, with a shift in time due the finite velocity of information propagation in the chain. We also observe a prolongation of the time the end qubit spends in its near-constant minimum of $Z$ value and an increase in this value – all of which appear (approximately) linear in $N$. These effects are approximately independent of the incoherent terms (at least for weak dissipation as here), and are the result of the excitations propagation with a fixed speed discussed in Sec. 3.2. On a longer timescale, the incoherent Lindbladian terms play an important role that determines the steady state, which we now turn to study.

The evolution of the second Rényi entropy $S_2$ [defined in Eq. (15)] is plotted in Fig. 5.[2] The second Rényi entropy quantifies the impurity of the density matrix, or its degree of being a mixed state rather than a pure state. In the absence of driving, the system would settle in the ground state (which is a pure state, with $S_2 = 0$) due to the Lindblad terms $g_0$ causing energy relaxation. The constant injection of excitations at qubit 0 modifies the steady state of the dynamics. Excitations are created on qubit 0 and propagate due to the couplings as discussed above, and at the same time are being destroyed by the loss terms, at some finite rate. Because of this decay the excitation density will decrease (exponentially) as we move away from the driven qubit, and far enough on the right the qubits will be exponentially close to their ground state. For this reason $S_2(t)$ should have a finite limit in the thermodynamic limit $N \to \infty$, coming from the contribution of a finite region near the driven qubit.

The value $g_0 = 0.1$ chosen above can be considered as a weak energy relaxation rate (as compared with the interaction timescale in Eq. (22) and the rotation speed given by $\Omega$). In App. D we study a range of $g_0$ values and present some examples of correlation functions in the steady state of the chain, showing that the two-qubit correlations decay with the distance between the qubits. In the next section we study an opposite example where large correlations form between distant qubits in the system, peaking on the two qubits farthest apart from each other.

---

[2]Note that although $S_2$ appears nearly converged (to better than a 1% level) at times $t \approx 20$, in fact some other observables (that oscillate) require somewhat longer simulation times (up to $t = 30$) to reach a similar level of convergence to their steady state values.

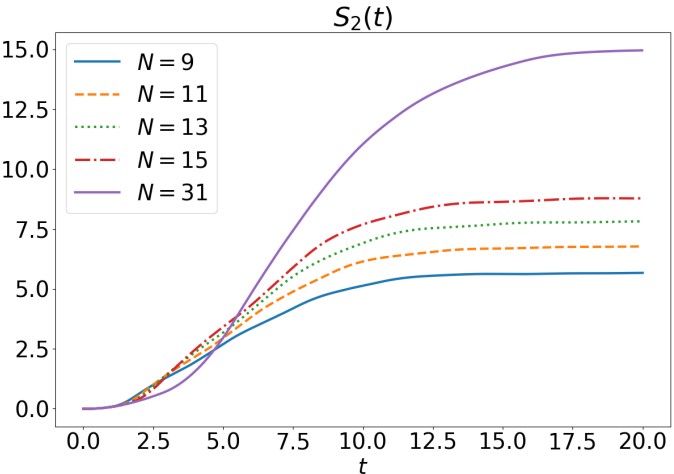

Figure 5: The second Rényi entropy $S_2$ [defined in Eq. (15)] vs. $t$, for the same simulations as presented in Fig. 4. The value of $S_2$ (which is essentially the log of the total purity loss in the system) is seen to converge to value apparently linearly decreasing with $N$.

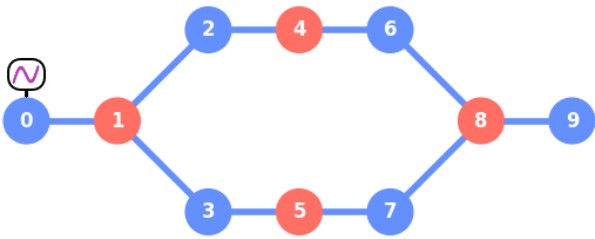

Figure 6: A plaquette with single edge qubits, containing ten qubits in total, alternating in frequency as depicted by their color. The bonds indicate qubits interacting with with their nearest neighbors by a 'flip-flop' (XY) interaction term as in Eq. (21). Qubit 0 is being driven periodically as in Eq. (20).

## 4 Alternating-frequency qubits in a plaquette topology

### 4.1 Dynamics of qubit excitations

We now turn to studying the dynamics of qubits with a lattice connectivity that goes beyond 1D. Motivated by the topology of current deployed IBM Quantum devices [45], we consider qubits in a configuration that we refer to as a plaquette. An example of such a topology is shown in Fig. 6 with 10 qubits. Without the edge qubits (qubits 0 and 9), the shown topology is a 1D ring, to which the edge qubits are connected at opposite points.

As mentioned in Sec. 2.5, in order to maintain a given precision in the calculations, the plaquette that comprises of a closed ring requires approximately the square of the site bond dimension, as compared with a similar simulation for the open chain. As a result, we find that simulations with model parameters similar to those of Sec. 3.3 are quite demanding numerically. In App. E we analyze this setup in some detail, and also present results indicating that the long dip in ⟨Z⟩ observed with 1D open chains disappears in the plaquette configuration.

Instead, we now consider qubits whose lab frequencies alternate, as illustrated in Fig. 6. There, the coloring of the qubits is in accordance with the on-site qubit frequencies as in

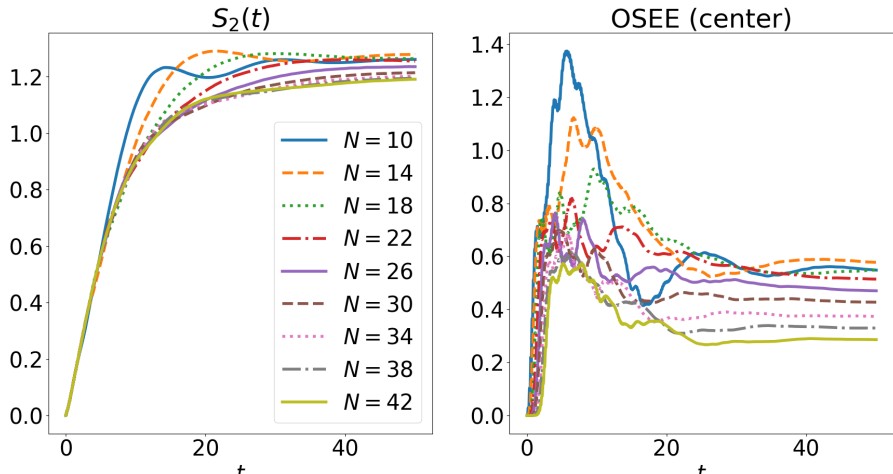

Figure 7: The second Rényi entropy $S_2$ (left), and the operator-space entanglement entropy for a bipartition of the system in its center (right), vs. the time $t$. The solutions are presented for $N$ between 10 and 42 in a configuration with alternating qubit frequencies similar to that of Fig. 6 (but extended), and the parameters are given in Eqs. (33)-(34).

Eq. (19) and Eq. (24). With the qubits no longer being resonant, the spread of correlations is slowed down (for a fixed interaction strength), and it turns out that a relatively low bond dimension is sufficient for capturing the system dynamics accurately, even when some long-ranged correlations (extending over the entire system) develop large values. In this section we set the parameters to be

$$\Omega = 2\pi \times 10, \qquad \Delta = 2\pi \times 5, \tag{33}$$

with the first qubit still being the driven one, and set the noise parameters being as above,

$$g_0 = 0.1, \qquad g_2 = 0. \tag{34}$$

The results presented in this subsection have been obtained using the simulation parameters determined/optimized in App. F.

As discussed for the linear chain in Sec. 3.3, when a drive term on a single qubit competes with an extensive number of loss terms ($g_0$) the steady state can have a finite density of excitations only in a *finite region* around the driven qubit. Far enough from the driven site the qubits are in the ground state. When the system size $N$ exceeds the size of this nontrivial region the entropy $S_2(t)$ no longer depends on $N$ (since a qubit in the ground state does not contribute to $S_2$). This is what is observed in the left panel of Fig. 7 where the curves for $N = 34, \ldots, 42$ are practically on top of each other. The excitation propagation is slower with this pattern of alternating frequencies (as compared with the resonant qubits of Sec. 3.3), so the loss terms are more effective in preventing the excitations from reaching qubits far from the driven one. The rapid convergence with $N$ displayed in the left panel of Fig. 7 indicates that about $\sim 30$ qubits are significantly affected by the drive.

The right panel of the same figure shows the OSEE defined in Eq. (17) for the same simulations. The OSEE computed for a bipartition in the center of the system measures the total amount of correlations between the qubits in the left and the in the right parts of the system. All correlations between these two subsystems contribute to the OSEE: the two-qubit correlations but also higher order ones, classical correlations as well as quantum ones.[3] Another

---

[3]The OSEE alone does not tell if the correlations are moslty classical or quantum.

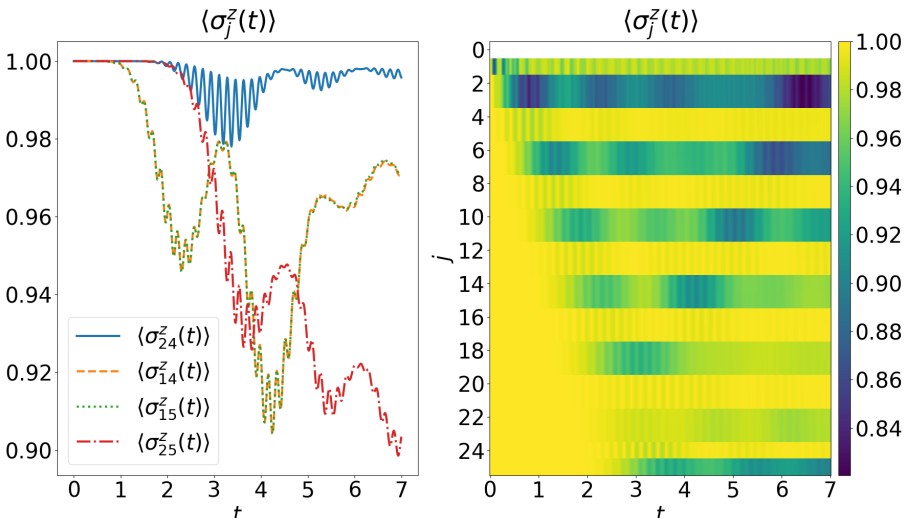

Figure 8: The *Z* Pauli expectation values vs. time for 26 qubits in a plaquette as in Fig. 7, for $t \leq 7$. Time curves are given for 4 qubits (left panel), and a space-time diagram for the whole system (right panel). The color bar on the right associates the scale of colors to the expectation value of each qubit (with qubit 0 excluded for clarity), plotted in rows from $j = 1$ to $j = 25$. The qubits that have a higher lab frequency (with $h_{z,i} = \Delta = 2\pi \times 5$) remain close to the ground state, while the ones with a lower lab frequency (with $h_{z,i} = 0$) develop a somewhat lower $\langle Z \rangle$ value. Two superimposed oscillation timescales are clearly visible, and pairs of qubits in the upper and lower arms of the plaquette (e.g., 14 and 15), evolve identically by symmetry – see the text for a detailed discussion.

important property is that the OSEE is closely related to the computational cost to store and evolve the state using an MPO representation [63]. Roughly speaking, for a fixed precision, the *logarithm* of the bond dimension (at the position of the bipartition) should be proportional to the OSEE. As can be seen in the right panel of Fig. 7 we start from a vanishing OSEE at time zero (the initial state is a product state without any correlations). The OSEE then grows due to the development of correlations between the two system parts as excitations propagate. After reaching some maximum the OSEE decreases as the dissipation gradually destroys some correlations. It becomes constant when approaching the steady state. It turns out that the qubits that have the strongest correlations are those in the vicinity of the driven qubit. Since the displayed OSEE corresponds to a cut in the center of the system, the larger the number $N$ of qubits the more distant is the bipartition cut from the strongly correlated region. This is a plausible explanation for the visible decrease of the height of the OSEE peak with the system size.

Fig. 8 shows $\langle Z \rangle$ using a space-time diagram for all qubits in the chain, except for qubit 0 that is excluded for clarity since its oscillations have a much larger magnitude. A light-cone of propagation is clearly visible, and is similar to the one in the previous setup (with interaction strength being identical in both). However, there is a significant difference due to the alternating qubit frequencies serving as effective off-resonant "blocks" along the chain. The higher-frequency qubits remain much less excited from the ground state. Also, visible throughout this section is the symmetry in the plaquette configuration, with the qubits in the upper and lower arms evolving identically under the effect of the driving. It should be noted that this symmetry is artificially broken by the choice of the MPO path used to represent the states in the simulations. The fact that this symmetry is nevertheless obeyed by the solution is a nontrivial indication of the accuracy of the simulation.

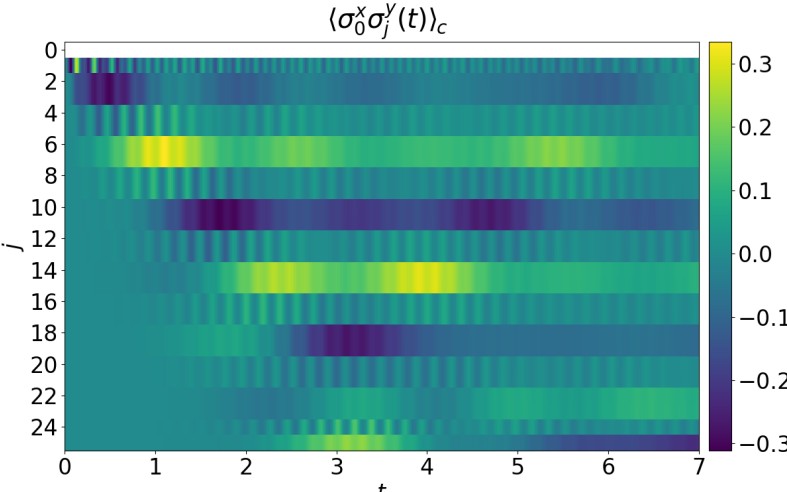

Figure 9: A space-time diagram of the $XY$ connected correlation function of qubit 0 with all of the other 25 qubits in the same simulation as in Fig. 8. Notable correlations develop between the Pauli-$X$ operator of the driven qubit and Pauli $Y$ of the qubits whose lab frequency is resonant with it, located at alternating positions along the two plaquette arms. These correlations build up following the propagation of the front of the information signal from the driven qubit, as can be seen by comparison with Fig. 8. The ensuing full two-point correlation matrix in the long-time limit is shown in Fig. 11.

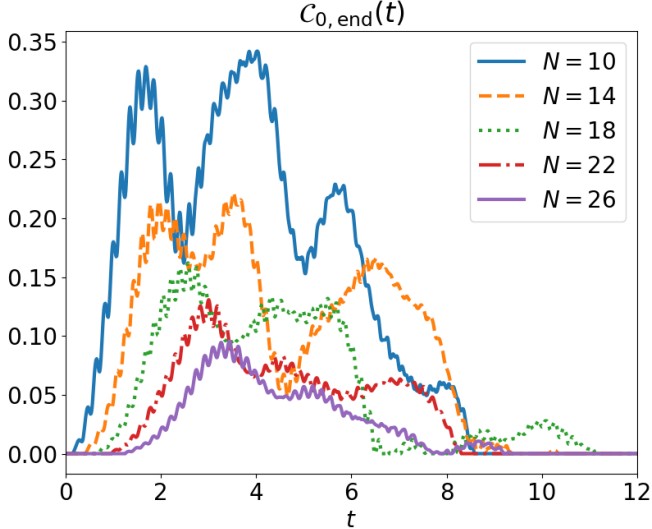

Figure 10: The entanglement concurrence $\mathcal{C}$ between the two edge qubits vs. the time $t$, for increasing system sizes in a plaquette topology. The parameters are given in Eqs. (33)-(34), with $N$ indicated in the plot legend for each curve. Rich non-monotonous entanglement dynamics can be observed for different systems sizes – the entanglement peaks together with the two-point correlation functions, decreases and increases again as travelling excitations reflect from the system boundaries, and eventually the entanglement vanishes even though some correlators remain relatively large in the steady state. See the text for a discussion.

## 4.2 Dynamics of correlation spreading

In contrast to the relatively weak excitation of the qubits due to the off-resonant blocking, we find that two-qubit correlations build up in the system and accumulate to a relatively large magnitudes between the two edge qubits of the plaquette. In the following we study connected correlation functions, defined for any time by

$$\left\langle \sigma_i^a \sigma_j^b \right\rangle_c \equiv \left\langle \sigma_i^a \sigma_j^b \right\rangle - \left\langle \sigma_i^a \right\rangle \left\langle \sigma_j^b \right\rangle. \tag{35}$$

The most notable correlator that we focus on here is the $XY$ connected correlation function, whose propagation is shown using a space-time diagram for qubit 0 with all other qubits in Fig. 9. Two timescales are visible both in this figure and in Fig. 8, which result from the fast driving (manifesting as a small-amplitude "micromotion"), and the slower timescale that determines the propagation of correlations along the system. In App. H we derive an effective expression for next-nearest neighbor interactions that mediate this correlation spreading. Eliminating the higher frequency qubits in the limit of $J \ll |\Delta|$, we find an XY interaction between the next-nearest qubits with $h_{z,i} = 0$, whose strength is

$$J_{\text{eff}} = J^2/|\Delta|. \tag{36}$$

In Fig. 10 we further show the dynamics of the concurrence (an entanglement monotone for two qubits in a general mixed state, [64]), calculated from the reduced density matrix of the two edge qubits (the driven one and the end one), for a few of the above presented system sizes. The entanglement clearly peaks with the arrival of the correlations at the end qubit (compare with Fig. 9 for the 26 qubits data, and Fig. 14 for $N = 22$). The maximal edge-qubits concurrence decreases monotonically with the system size, which is plausible due to the

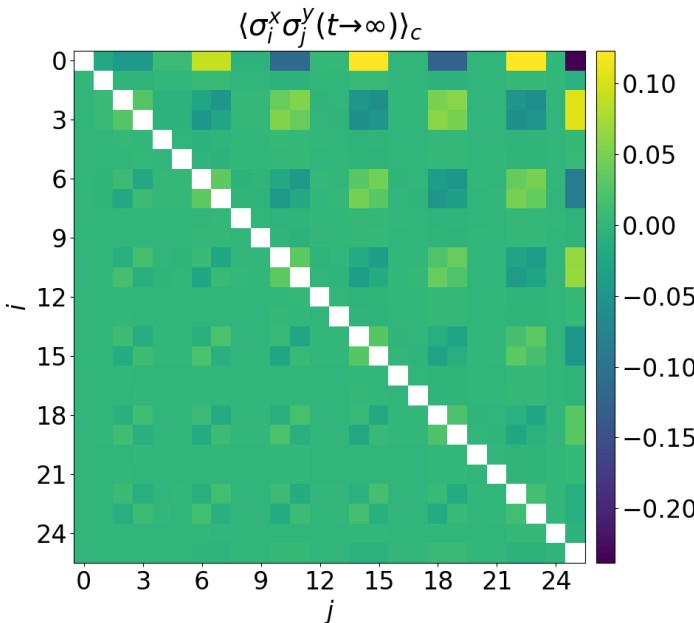

Figure 11: The $XY$ connected correlation matrix of all qubit pairs at the final time, in the same simulation as in Fig. 8. This two-point correlation function appears to *increase* (in magnitude) with the distance between the two qubits in the plaquette. The largest correlation function is between the two edge qubits – qubit 0 on the left and qubit 25 on the right.

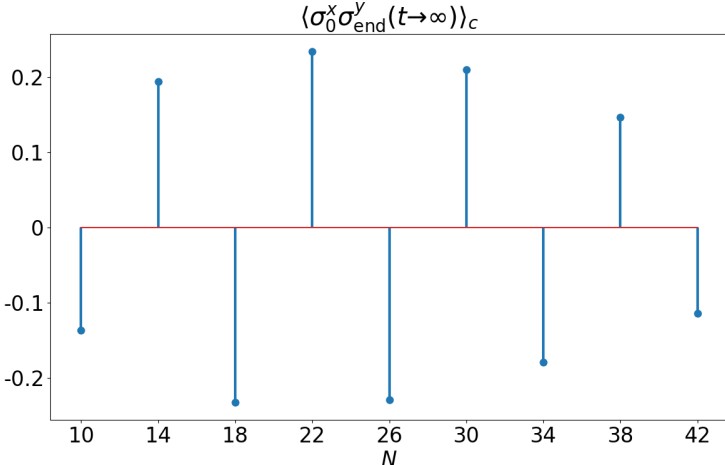

Figure 12: The value of the $XY$ connected correlation function of qubit 0 with the end qubit as a function of $N$ in the same simulations as in Fig. 7. The initial increase (with $N$) and the eventual decay are the plausibly result of competition between the driving that creates excitations, their dynamics of propagation and reflections in the system, and the continuous energy relaxation – see the text for a discussion.

larger distance and competing processes. Following the initial peak of the concurrence it decreases and increases again, which can be attributed to reflections from the system boundaries. Eventually the incoherent relaxation sets in and consistent with the characteristic fragility of entanglement referred to as "entanglement sudden death" [65], beyond a certain time the concurrence reaches zero and (after a possible transient resurgence most clearly seen with $N = 18$), stays null.

However, although the entanglement of the edge-qubits in the long-time limit is zero, the qubits remain correlated in the steady state that becomes nontrivial. Through the effective next-nearest interaction that allows correlations to build up across the off-resonant barriers, we find that in the approach to the steady state the connected correlation function unexpectedly increases in magnitude with the distance between qubits, as shown in Fig. 11. This is most pronounced on the first row and last column of the figure, with the $XY$ correlation peaking between the two edge qubits. In Fig. 12 the $XY$ correlation between the two edge qubits in the steady state is shown as a function of $N$ for the same simulated systems as in Fig. 7. Starting from $N = 10$, the magnitude of the correlator increases with $N$, peaks at $N = 22$, and for larger systems decreases again. It is determined by a competition between the driving term and the incoherent relaxation, in combination with the qubit parameters and connectivity.

## 4.3 Robustness of the correlations

Although it is beyond the scope of the current study to analyze and model in more detail the correlation dynamics pointed at in the previous subsection, some further simulations suggest that these nonlocal correlations are robust when parameters such as driving amplitude and qubit connectivity are modified. Moreover, we consider the effect of dephasing ($g_2$) introduced into the problem, and also small ZZ coupling terms in addition to the XY terms. The ZZ terms open a new channel for interaction that is not directly suppressed by the off-resonance condition. This leads to an increase in the complexity of the simulations as evidenced by global quantifiers, and for large enough $J_z$ values results in the disappearance of the nonlocal correlations observed above. As we discuss in more detail in the next section, superconducting qubit devices indeed often have small ZZ coupling terms whose mitigation and control form one of

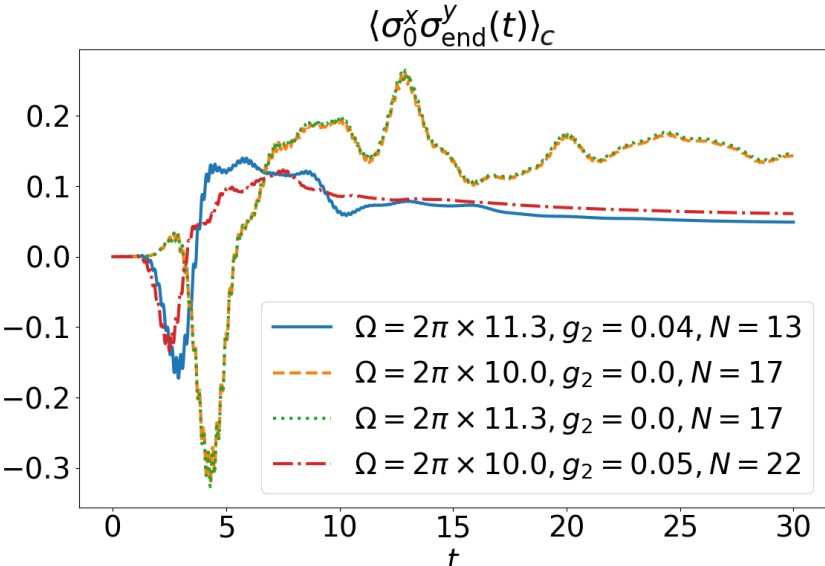

Figure 13: The two-qubit $XY$ connected correlation function of the two edge qubits vs. the time $t$. The curves with $N = 13, 17$ correspond to a linear chain configuration, while $N = 22$ is for a plaquette topology, where the qubit frequencies alternate between nearest neighbors in both cases. The parameters are given in Eq. (37), in addition to the plot legend for each curve. The relatively large two-point correlations between the edge qubits are robust to variations in the drive amplitude and can be seen in both topologies studied in this work. For the shown (moderate) dephasing strengths, these correlations appear to remain nonnegligible, although reduced in amplitude as can be expected.

the outstanding challenges in the field. The possibility of simulating realistic multiqubit dynamics in parameter regimes corresponding to actual quantum hardware remains a promising extension of the current work.

In Fig. 13 we show the dynamics of the $XY$ correlation function between the two edge qubits, in several configurations with different parameters, where the qubits have alternating frequencies. The plot shows data for 13 and 17 qubits in a linear chain configuration, and also in a plaquette configuration with 22 qubits, as studied in Sec. 4. The parameters for these simulations, using the definitions in Sec. 3.1 are

$$\Delta = 2\pi \times 5, \qquad g_0 = 0.1, \tag{37}$$

and $J = 2\pi \times 1$ as in Eq. (22), with $\Omega$, $g_2$ and $N$ specified in the plot legend for each curve. The figure shows that the relatively large two-point correlations between the edge qubits are not particular to the drive amplitude or the plaquette topology studied in Sec. 4. For the dephasing strengths of $g_2 = 0.04, 0.05$, the correlations are reduced in magnitude as could be expected, but do not disappear completely. We may therefore expect that they are robust – at least to some extent – to more general parameter variations.

Indeed, in Fig. 14 we present for comparison the edge-qubits $XY$ correlation with a similar model but including an additional $ZZ$ (Ising) coupling, corresponding to the coupling coefficient $J_z \equiv J_{ij}^z$ of Eq. (4) between neighboring qubits. The parameters are

$$\Delta = 2\pi \times 5, \qquad \Omega = 2\pi \times 10, \qquad J = 2\pi \times 1, \qquad g_0 = 0.1, \qquad g_2 = 0, \tag{38}$$

and $J_z$ specified in the plot legend for each curve. The ZZ coupling added to the XY model induces a further perturbation taking the setup away from the limit of a model with integrable

dynamics in the bulk. As Fig. 14 shows, the long-range correlations established in the system wash away progressively as this term is increased, which appears to go hand in hand with an increased complexity of the dynamics, as indicated by the larger values of $S_2$ and the OSEE shown in Fig. 15.

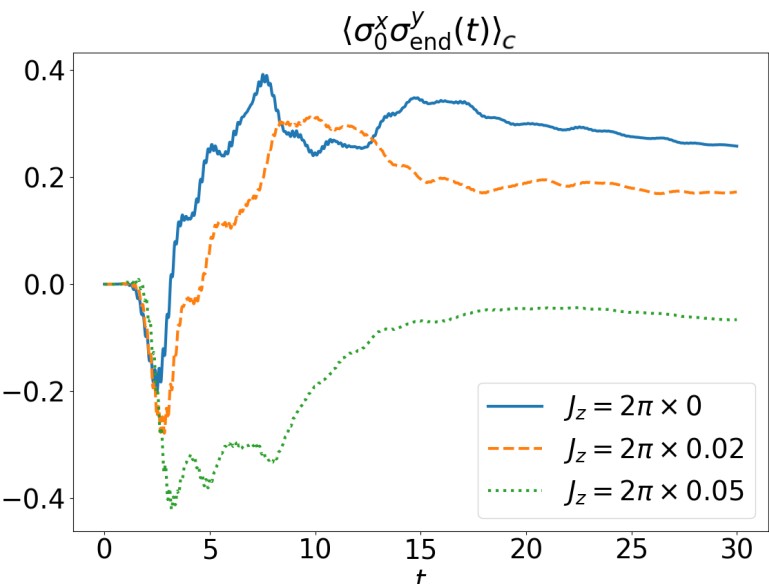

Figure 14: The two-qubit $XY$ connected correlation function of the two edge qubits vs. the time $t$, with $N = 22$ for a plaquette topology where the qubit frequencies alternate between nearest neighbors, which also coupled via ZZ interaction. The parameters are given in Eq. (38), with $J_z$ indicated in the legend. As the ZZ coupling strength is increased, the absolute value of the long-time correlation is gradually reduced.

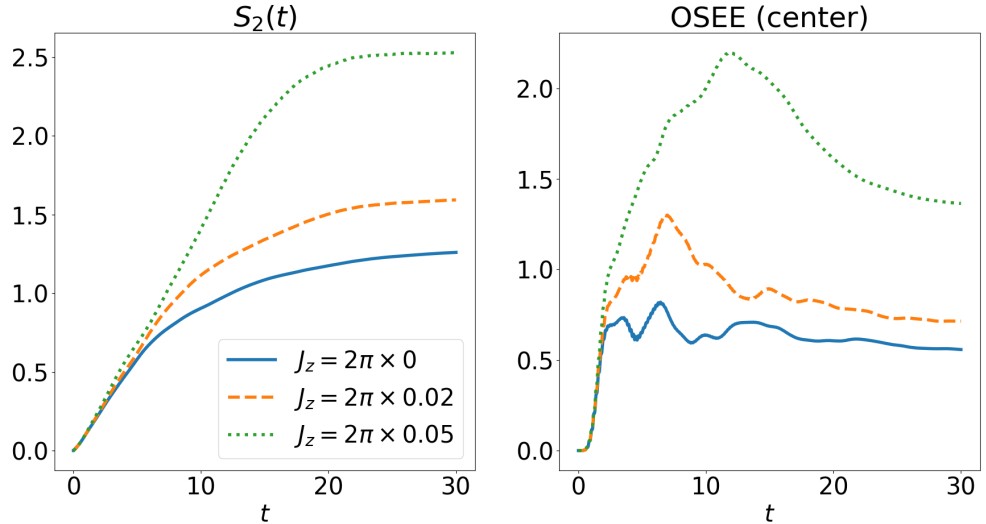

Figure 15: The second Rényi entropy $S_2$ (left), and the operator-space entanglement entropy for a bipartition of the system in its center (right), vs. the time $t$. The simulations are identical to those in Fig. 14 ($N = 22$ plaquette topology), showing a clear increase in the complexity of the dynamics with increasing $J_z$.

# 5 Summary and Outlook

In this work we introduced a numerical solver for Lindblad dynamics that is capable of evolving the density matrix of tens of qubits in some physically relevant setups. Because it uses an MPO representations it is particularly adapted for systems where the qubits connected in 1D or quasi 1D geometries, but it can also be used in more general setups. The computational cost of a simulation depends on the amount of correlations in the system, as quantified by the OSEE.

The solver consists of a high performance C++ core and a feature-rich Python interface. In addition to a comprehensive documentation and some tutorials, the source code used for generating the research presented in this work is available as an example within the solver repository [43], and can be used as a helpful starting point for conducting a research project managing hundreds of simulations using a local dataframe in Python. Using single-qubit and two-qubit observables chosen according to symmetries of the solution, in addition to global entropy quantities, we benchmarked the required numerical parameters for achieving reliable results with optimal performance. We compared the results of the simulations to essentially exact independent numerical simulations and earlier literature. As referenced in the sections above, the Appendix presents the details of this investigation, which would be useful for future studies employing the current solver in numerically challenging parameter regimes.

Studying a setup where an edge qubit in an XY model with nearest neighbor interactions is driven resonantly, we analyzed two main configurations. The first one is composed of identical qubits and the second one has qubits whose frequencies alternate between neighbors. In both cases, we have considered a linear chain and a plaquette with an enclosed ring. We have covered different parameter regimes in terms of the dynamical timescales of the Hamiltonian and in terms of the incoherent rates. For identical qubits in a linear chain (with open boundary conditions) we found a few interesting properties. The first one is the $-\pi/2$ angle difference between neighboring qubits. The second one is the relatively long "dip" in $\langle Z \rangle$ of the right-edge qubit, for a duration that appears to grow linearly with the system size, stable until excitations have enough time to travel back and forth between the configuration edges. The competition of the dynamical timescales and energy relaxation (together with edge effects leading to reflections of travelling excitations) determine inhomogeneous spatial patterns in the steady state reached for various parameters.

Our main result concerns the simulations of qubits with alternating frequencies between nearest neighbors, in particular in a plaquette configuration. Although neighboring qubits interact weakly due to the off-resonance condition, the excitations generated by the drive propagate through the lattice and lead to large correlations with the driven qubit. In contrast to the typical cases where correlation functions decay with distance, we have found here that the correlations increase with the distance between qubits. In the steady state, the two edge qubits – in both the linear chain and the plaquette – develop the largest two-point correlations.[4] It is beyond the scope of the current work to characterize completely how this depends on all the parameters, however, this emerging correlation between distant qubits is of interest in particular in qubit devices and quantum computation, which routinely involve high-frequency resonant qubit driving to perform quantum gates. Qubits far from the driven qubit are typically assumed to remain uncorrelated with it, and hence mechanisms that result in nonlocal correlations are important to identify.

The simulations of alternating-frequency qubits in a plaquette are motivated by current deployed IBM Quantum devices, which realize a topology comparable to a heavy-hexagonal connectivity, with prospects for a designated quantum error correction code. In some of those devices the XY interaction coefficient between connected qubits is of order $J \sim 2\pi \times 2\,\mathrm{MHz}$. Fix-

---

[4]In particular in the $XY$ correlation function that expresses $\pi/2$ phase differences in the $xy$ plane of single-qubit Bloch spheres

ing this scale for the parameters and considering a frame rotating at a typical qubit frequency, values for $h_{z,i}$ (expressing qubit frequency differences) are about an order of magnitude larger as compared with the parameters taken in our simulations. The incoherent terms are currently roughly an order of magnitude smaller than those we have considered, and the ZZ coefficients are roughly of order $J_z \sim 0.02J$, within the range studied in App. 4.3. In those superconducting qubit devices, the coefficients vary between the qubits (within some bandwidth), and the system can be considered as disordered or inhomogeneous. This is in contrast, e.g., to cold atoms [66] and trapped ions [67] where the qubits are identical (in the former case the coupling is short-ranged and uniform, while in the latter typically long-ranged and varying).

Therefore, our study can be considered as a first step towards the simulation of dynamics with realistic device connectivity and qubit parameters. Interesting extensions could be to consider $d$-level qubit dynamics (suitable for transmon qubits for example), with more general Hamiltonian and Lindbladian parameters, and time-dependence in the parameters allowing to directly integrate more complex driving protocols. Despite possible notable other differences (e.g., non-Markovian effects with superconducting qubits [68]), it is an interesting and open question to see whether effects observed in this work remain relevant to some extent, and what new effects could be simulated and possibly observed in experiments with those devices. In particular, nonlocal correlations forming as a result of qubit driving such as those presented in our results could be of fundamental importance when considering quantum error correction codes on multiqubit devices.

## Acknowledgements

We thank Gal Shmulovitz and Eldor Fadida for significant contributions to the Python source code of the solver, and Dekel Meirom for significant contributions to its documentation. H.L. thanks Gadi Aleksandrowicz, Eli Arbel, Daniel Puzzuoli, Chris Wood, Matthew Treinish, Moshe Goldstein, Noa Feldman, and Matan Lotem for very helpful feedback. Research by H.L. was partially sponsored by the Army Research Office and was accomplished under Grant Number W911NF-21-1-0002. The views and conclusions contained in this document are those of the authors and should not be interpreted as representing the official policies, either expressed or implied, of the Army Research Office or the U.S. Government. The U.S. Government is authorized to reproduce and distribute reprints for Government purposes notwithstanding any copyright notation herein. G.M. is supported by the PEPR integrated project EPiQ ANR-22-PETQ-0007 part of Plan France 2030.

## A  Basics of MPS and MPO

An MPS [25] is a particular way to encode a many-body wave-function using a set of matrices. Consider a system made of $N$ qubits, in a pure state

$$|\psi\rangle = \sum_{s_1, s_2, \cdots, s_N} \psi(s_1, s_2, \cdots, s_N) |s_1\rangle |s_2\rangle \cdots |s_N\rangle. \tag{39}$$

In this expression the sum runs over the $2^N$ basis states ($s_i \in \{0, 1\}$) and the wave-function is encoded into the function

$$\psi : \{s_i\} \rightarrow \psi(s_1, s_2, \cdots, s_N). \tag{40}$$

An MPS is a state where the wave function is written

$$\psi(s_1, s_2, \cdots, s_N) = \text{Tr}\left[ A_1^{(s_1)} A_2^{(s_2)} \cdots A_N^{(s_N)} \right], \tag{41}$$

where, for each qubit $i$ we have introduced two matrices $A_i^{(0)}$ and $A_i^{(1)}$ (for a local Hilbert space of dimension $d$ one needs $d$ matrices $A_i^{(0)}, \cdots, A_i^{(d)}$ for each qubit). These matrices are in general rectangular and the wave-function is obtained by multiplying them, as in Eq. 41. What determines the dimensions of the matrices? If the matrices are one-dimensional (scalar), one has a trivial product state (and all product states can be written this way). On the other hand, if one allows for very large matrices, of size $2^N$, any arbitrary state can be written as an MPS. In fact the MPS representation is really useful when the system has a moderate amount of bipartite entanglement. As a "rule of thumb", to get a good MPS approximation of a given state, each matrix $A_i^{(s_i)}$, of size $d_{i-1} \times d_i$, should have a dimension $d_i$ of the order of $e^{\text{const.} \times S_{\text{vN}}(i)}$, where $S_{\text{vN}}(i)$ the von Neumann entropy of the subsystem $[i+1, \cdots, N]$.

What about *mixed* states? They can be represented using so-called matrix-product operators (MPO):

$$\rho = \sum_{a_1, a_2, \cdots, a_N} \text{Tr}\left[ M_1^{(a_1)} M_2^{(a_2)} \cdots M_N^{(a_N)} \right] \sigma_1^a \otimes \sigma_2^a \otimes \cdots \sigma_N^a , \qquad (42)$$

where each $a_i$ can take four values $\in \{1, x, y, z\}$, $\sigma^{a_i}$ is a Pauli matrix or the identity acting on qubit $i$, and we have associated four matrices $M_i^{(1)}$, $M_i^{(x)}$, $M_i^{(y)}$ and $M_i^{(z)}$ to each qubit.

The MPO representation allows to encode any operator, not only density matrices. In particular this representation does not enforce the positivity or the Hermitian character of $\rho$ and numerical errors (due to the SVD truncations or finite time steps) can cause slight violations of these properties. A different representation, called matrix-product density operator (MPDO) [30] enforces these properties. We plan to include some MPDO support in some next version of this solver.

## B  Finding optimal simulation parameters for a chain of identical qubits

The two most important parameters controlling the numerical accuracy of the solver are probably the maximal bond dimension ($\eta$, explained below) and the discrete fixed time step of the simulation ($\tau$). The former parameter limits the amount of correlation (the OSEE in fact) between partitions of the chain which can be faithfully described by the solution ansatz, while the latter controls the Trotterization error. The computation time increases roughly with the third power of the bond dimension, and linearly in the time step $\tau$ (though the dependency is not always so simple). We discuss below also other relevant numerical parameters.

In the current solver the density matrix of $N$ qubits is represented as a pure state in the Hilbert space of $N$ four-level systems. In this pure state the Schmidt decomposition of the reduced density matrix of any of the two sub-systems of any bipartition of the qubits would have at most $4^{\lfloor N/2 \rfloor}$ singular eigenvalues. Hence, this is the maximal bond dimension required for a numerically exact solution in the presented simulations. Our strategy for fixing optimal parameters for large system simulations is based on starting with a system of up to 11 qubits. We vary different numerical parameters, comparing the results and verifying their convergence. The maximal bond dimension would not exceed 1024 in this case, and for such qubit numbers also a brute-force matrix solution is available, allowing us to compare two independent numerical solutions of the dynamics.

Using this approach, we have tested different values for $\tau$ and the bond dimension. We set the Trotterization order to 4 throughout this work. With the fastest timescale set by the interaction $J$ of Eq. (22) multiplied by the number of the nearest neighbors a qubit has, we tested different $\tau$ values between $\tau = 0.005$ and $\tau = 0.05$. With $\tau = 0.05$ we find that the time evolution steps begin at some point to introduce noticeable errors that manifest as a non-

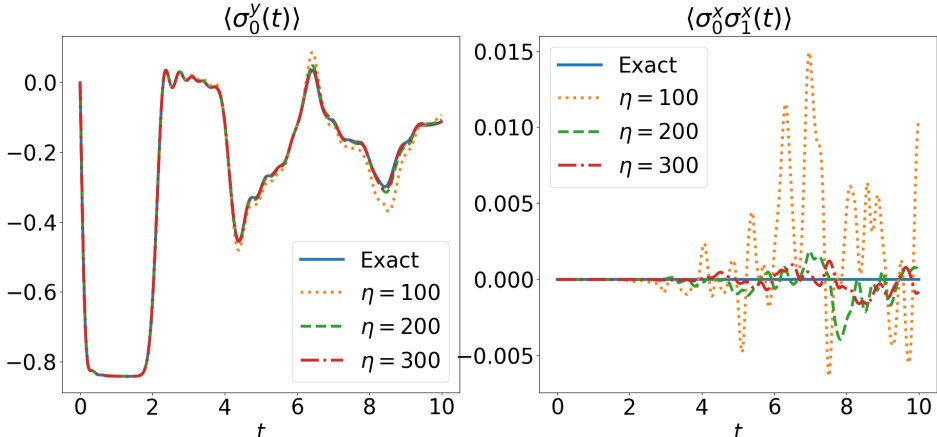

Figure 16: The $Y$ expectation value for qubit 0 (left), and the two-qubit $XX$ expectation value for qubits 0 and 1 (right), for $N = 11$ qubits with energy relaxation parameter $g_0 = 0.1$ as in Eq. (32), and the other parameters as in Figs. 2-3. A few solutions are shown vs. the time $t$, simulated by using both a numerically exact simulation (storing the full density matrix), and the MPO approximated simulation with different maximal bond dimension $\eta$, with the convergence towards the exact solution with increasing $\eta$ is clearly seen.

Hermitian density matrix and imaginary components in expectation values that are $> 10^{-4}$. The deviations from a Hermitian $\rho$ can be compensated by effecting a forced Hermitization at the end of each step, assigning $\rho \rightarrow \left(\rho + \rho^{\dagger}\right)/2$ (a step that can be done also intermittently), however inaccuracies in the solution can obviously still accumulate. For most of the below simulations we have used $\tau = 0.02$ (or $\tau = 0.03$ in some cases, which comes at a small price in precision), and only forced $\rho$ to be Hermitian every tenth time step. The density matrix is further forced to remain valid at each time step by effecting $\rho \rightarrow \rho/\text{tr}\{\rho\}$.

We also tested some values of the cutoff parameter. After a one step evolution the bond dimension of the MPO which represents $\rho$ increases. As in any MPS or MPO-based algorithm, some compression needs to be done to keep the bond dimension under control. This is done (inside the iTensor library) at the level of the vectorized state $|\rho\rangle\rangle$, by computing the Schmidt spectrum associated to each bond, and discarding the singular values that are beyond some threshold. The actual truncation is done using the most severe condition between `cut_off_rho` (which sets the maximum weight that can be discarded) and `max_dim_rho`. Our experience is that the best results are obtained by keeping `cut_off_rho` very low, at a value of $1e-14$ or $1e-16$. When starting the simulation from a product state this insures that a large number of Schmidt values are kept at the early stages of the evolution, and that the truncation only starts when the bond dimension reaches `max_dim_rho`.

Fixing all other parameters, Fig. 16 shows the convergence towards the numerically exact values of 1Q and 2Q observables as a function of the maximal allowed bond dimension from $\eta = 100$ to $\eta = 300$. The $Y$ expectation value of qubit 0 (left), and the $XX$ expectation value of qubits 0 and 1 (right) are shown as a function of the time up to $t = 1/g_0 = 10$, a time characteristic of the dynamics of growth of correlations in the chain. With $\eta = 100$ the value of $\langle\sigma_0^y\rangle$ shows noticeable deviations, reaching even the order of 10% near a turning point of the curve (although it continues to follow the dynamics). This observable clearly converges with $\eta = 200, 300$ towards the exact value. It is typically important to verify not only single qubit observables but also two-qubit observables and global density matrix quantities, which may be more sensitive to simulation errors. Here we suffice with examining $\langle\sigma_0^x\sigma_1^x\rangle$, whose maximal error is $\approx 0.015$ for $\eta = 100$, and decreases by an order of magnitude for $\eta = 300$.

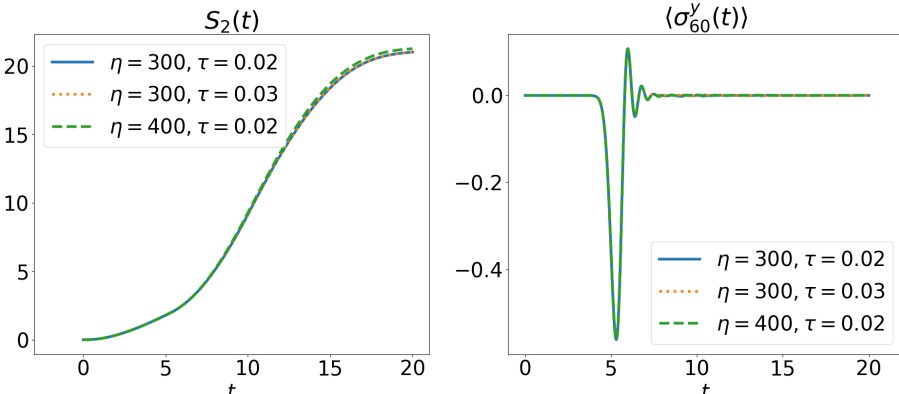

Figure 17: The second Rényi entropy $S_2$ (left), and the $Y$ expectation value for qubit 60 (right), for $N = 61$ qubits with energy relaxation parameter $g_0 = 0.1$ as in Eq. (32), and the other parameters as in Figs. 2-3. A few solutions are shown vs. the time $t$, simulated with different values for the maximal bond dimension $\eta$ and time step $\tau$, with the convergence clearly seen.

We have also studied the error in the second Rényi entropy $S_2$ (not shown), which leads to similar conclusions. In Fig. 17 we present $S_2$ as a function of the bond dimension in analysis of some simulations for $N = 61$ qubits, confirming the same accuracy.

With these conclusions, we can simulate long qubit chains that cannot be simulated by brute-force methods, with optimally chosen parameters. We note that we continue to monitor the accuracy of those simulations, and based on the above analysis, we can use two approaches. The obvious one is repeating simulations with increasing $\eta$ and decreasing $\tau$ and verifying that local observables and global quantities are converged. In addition, an even simpler but highly effective check would be to monitor quantities whose values are known from symmetries or found to obey certain patterns as with the alternating $xy$ projections of the qubits in the current problem.

## C  A dissipative quantum Ising ring

As a further example, we look at the convergence of the local expectation values $\langle X \rangle$, $\langle Y \rangle$ and $\langle Z \rangle$ in the steady state of a strongly dissipated quantum Ising ring. The model and parameters are the same as those previously considered for $N = 16$ in Ref. [69] with a different numerical method, which allows us to check our results. The Hamiltonian is characterized by $J_z \equiv J_{ij}^z = 1$ of Eq. (4), with a uniform transverse field (on all qubits) whose strength $h_x$ is varied from 0.1 to 5 (see Fig. 18). The dissipation comes from Lindblad operators $\sigma^-$ acting on all qubits, with strength fixed to $g_1 = 1$. Two systems with 16 and 32 qubits arranged in a ring (a chain with periodic boundary conditions) are considered.

The data plotted in Fig. 18 correspond to a final time $t_f = 20$, giving a good approximations to the steady state values. The maximal bond dimension was varied from $\eta = 50$ to $\eta = 200$ and two values of the time step $\tau$ were considered (0.05 and 0.1). In addition, the expectation values of the three components of the magnetization were verified to be translation invariant at the scale of the plot, which is an important check in presence of periodic boundary conditions. All these data show that the results are already well converged for a bond dimension $\eta = 50$, and can be seen to reproduce the results of [69]. We note also that the OSEE never exceeds $\sim 0.3$, so in this relatively strong bulk dissipation regime the MPO method could easily be applied to much longer Ising chains. The fact that the data for $N = 16$ and 32 are essentially

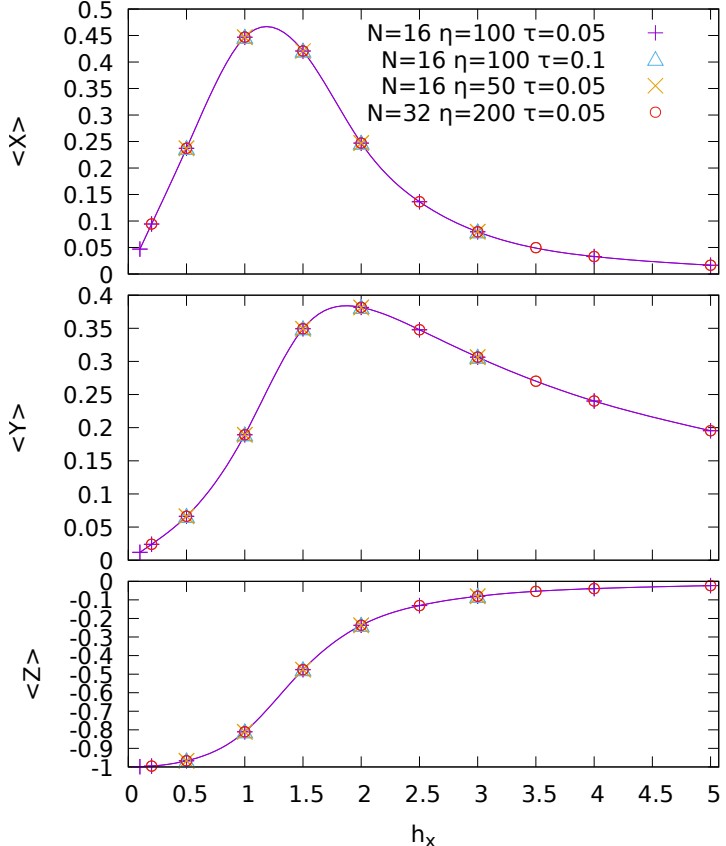

Figure 18: Expectation values $\langle X \rangle$, $\langle Y \rangle$ and $\langle Z \rangle$ in the steady state of a dissipative quantum Ising ring as a function of $h_x$, with $J_z = 1$, $g_1 = 1$ and $N = 16, 32$ qubits. The simulation parameters $\eta$ (maximum bond dimension) and $\tau$ (time step) are varied, showing the very good convergence of the results. The final time is $t_f = 20$ and cut_off_rho $= 10^{-16}$. The coincidence of the values for $N = 16$ and $32$ indicates that the correlation length is short. The line is a guide to the eye.

identical also indicates that the correlation length is relatively short. This short correlation length, low OSEE value, and low $\eta$ value for which convergence is observed (even with periodic boundary conditions) are plausibly the result of the large decay rate.

## D  Correlations in the steady state of a chain of identical qubits

In Fig. 19 we depict the expectation values of the single-qubit Bloch vector components in the steady state of 31 qubits with parameters as studied in Sec. 3.3, with increasing $g_0$ values, between $g_0 = 0.1$ and $g_0 = 1$. The $\langle Z \rangle$ steady state value of the qubits is seen to increase with $g_0$, as expected when increasing the strength of the relaxation. The driven qubit (number 0) has the largest expectation value in the $xy$ plane (pointing along $-y$), which is accompanied by a lower $\langle Z \rangle$ value, and the $xy$ plane projection is seen to decrease along the chain starting from qubit 0. The end qubit on the right of the chain also has a low $\langle Z \rangle$ value, similar to the first qubit, although this does not come with a notable $xy$ component.

The steady-state patterns shown in Fig. 19 emerge from the competition between the energy relaxation and the excitations propagating and reflecting back and forth in the chain. The dynamics at low $g_0$ appear to be overall underdamped, where multiple reflections along the

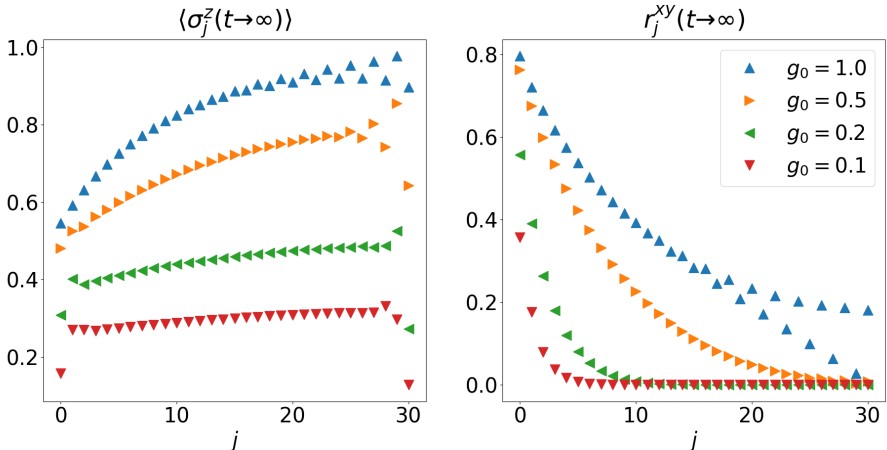

Figure 19: The expectation value of each qubit's Pauli components in the steady state, for $Z$ (left) and $X$, $Y$ (right), given as the length of the Bloch vector component in $xy$ plane, $r^{xy} \equiv \sqrt{\langle\sigma^x\rangle^2 + \langle\sigma^y\rangle^2}$, for 31 qubits in a 1D chain, with varying values of $g_0$, between $g_0 = 0.1$ and $g_0 = 1$, with the other parameters as in Fig. 4. The ending time of each simulation obeys $t \gg 1/g_0$, with the steady state approached faster for stronger decay rates. The competition of the energy relaxation with the excitation dynamics leads to the shown patterns – see the text for a detailed discussion.

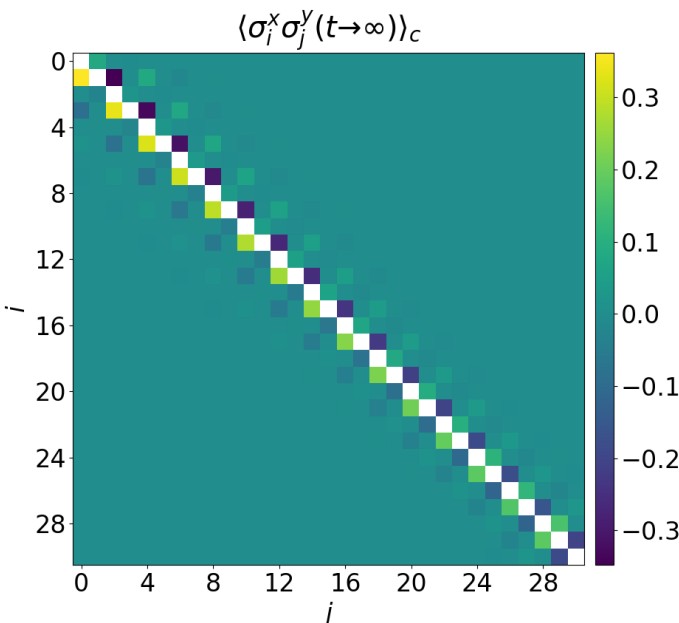

Figure 20: The matrix of $XY$ connected correlation function of all qubit pairs at the final time, in the simulation of 31 identical qubits in a 1D chain with $g_0 = 0.1$ shown in Fig. 19. Neighboring qubits close to the left edge have relatively large correlation functions, while correlations between qubits on the right side are weaker, and correlations clearly decay rapidly with distance throughout the chain.

chain plausibly lead to a relatively uniform $\langle Z \rangle$ pattern in the bulk with edge effects extending over just 1-2 sites. In the $X, Y$ expectation values, there is a simple exponential decay from the left edge (with a decay length increasing with $g_0$). For large enough $g_0$, edge effects extend into the bulk with a much larger penetration length, and a nonuniform alternating pattern of

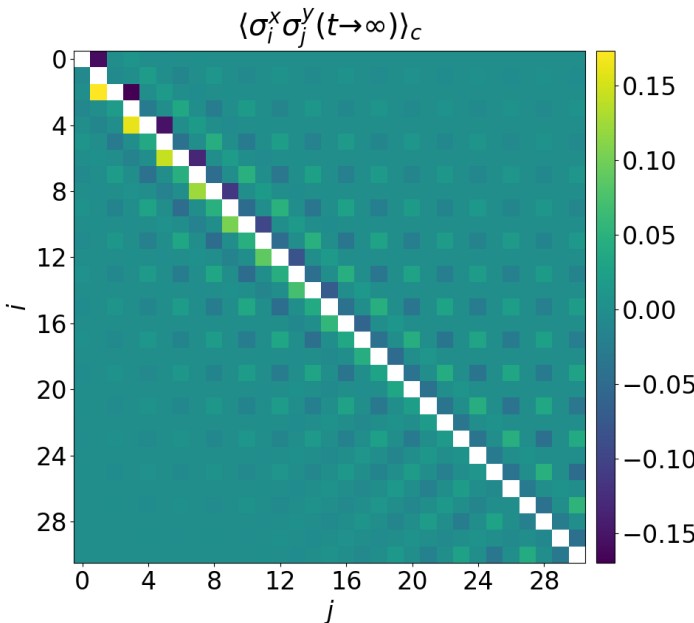

Figure 21: The matrix of $XY$ connected correlation function of all qubit pairs at the final time, similar to Fig. 20, but for the simulation with $g_0 = 1$ in Fig. 19. The correlations are smaller in magnitude as compared with Fig. 20, while their spatial decay length is noticeably longer.

on-site qubit orientations on the Bloch sphere, apparently "frozen" by the strong relaxation.

Figures 20 and 21 depict two $XY$ correlation matrices at final time of two simulations of 31 identical qubits in a 1D chain with $g_0 = 0.1$ and $g_0 = 1$, shown in Fig. 19. The system is close to its steady state in both cases, and for $g_0 = 1$ the qubits are globally closer to the ground state than for $g_0 = 0.1$. For this reason there are less degrees of freedom, or fewer excited qubits, available to develop some $XY$ correlations at $g_0 = 1$ than for $g_0 = 0.1$. The amplitude of the correlations are indeed smaller in the case where the losses are the strongest (Fig. 21) compared to Fig. 20. On the other hand, the correlation decay length is noticeably shorter (with an extension of 1-2 qubits) when $g_0 = 0.1$. This may be linked to the fact that the steady state at $g_0 = 0.1$ has a higher density of excitations than at $g_0 = 1$, and their mean spacing between excitations is therefore shorter. This mean spacing is a natural length in the problem which, in turn, affects the correlation length.

## E   Identical qubits in a plaquette

We now examine how the dynamics of identical qubits in a 1D chain with open boundary conditions change when the bulk of the chain is split to form a plaquette. An example of such a topology is shown in Fig. 22 with 10 qubits, in a configuration that we refer to as a plaquette. Without the first and last qubits (qubits 0 and 9), the shown topology is a 1D ring (i.e., the boundary conditions are periodic), and the edge qubit are connected at opposite qubits of the ring. The first qubit is still the one being driven, and we plot in the left panel of Fig. 23 the value of $\langle Z \rangle$ for the end qubit with plaquettes of increasing qubit numbers, up to $N = 22$. The relatively long "dip" in $\langle Z \rangle$ that was observed with 1D open chains has disappeared in the plaquette.

As mentioned in the introduction, the plaquette that involves closed loops, requires a squar-

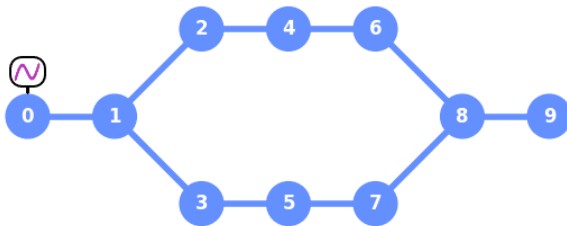

Figure 22: A plaquette with two edge qubits attached on opposite sides of a ring, containing ten qubits in total. The bonds indicate qubits interacting with with their nearest neighbors by a "flip-flop" (XY) interaction term as in Eq. (21). Qubit 0 is being driven periodically as in Eq. (20).

ing of the maximal bond dimension (in order of magnitude estimation) as compared with a simulation of similar dynamics in a 1D open chain to keep a given precision. The reason can be understood from the ordering of the qubits in Fig. 22. The qubit ordering indicates the order of local Hilbert spaces in the MPS representation, which is 1D by construction. As is customary and useful, we order the opposite-facing qubits on the two arms of the plaquette according to a "ladder" or "zigzag" that progresses from left to right uniformly. The maximal jump in index along the chain between interacting qubits is 2, and one can think of the configuration as comprising of unit cells of a double size. Since in the subsection App. B we saw that the maximal bond dimension parameter must be set to at least $\eta = 100$ for reasonable results (and in practice we have used $\eta = 300$ in most simulations), we may expect that $\eta \sim 10^4$ is required for simulating the same dynamics on the plaquette.

Indeed, limiting the simulations by setting $\eta = 1000$ (which is already quite demanding in terms of run time), we find that the dynamics on the plaquette of identical qubits can be simulated only for $t \lesssim 2$. In the right panel of Fig. 23 the Rényi entropy per qubit is plotted for those simulations with different numbers of qubits from $N = 10$ through $N = 22$. The simulation with $N = 10$ is numerically exact, and $S_2$ increases monotonously and sharply. For higher qubit numbers we see at $t \approx 1.7$ an inflection point in $S_2$, signalling for this setup that the simulation begins to break. Indeed, other observables (not shown) increasingly break symmetries of the system. The plaquette with qubit 0 obeys a symmetry between its upper and lower arm qubits (while the MPS ansatz does not), and observables must coincide on pairs of qubits such $(2, 3)$, $(4, 5)$, etc. Comparing expectation values on these pairs is an efficient method to detect the breakdown of the numerics.

## F Optimal parameters for alternating-frequency qubits in a plaquette

With the high frequency parameters set in Eq. (33), we must decrease the integration time step $\tau$, and we find that $\tau = 0.01$ is in fact sufficient to capture properly the faster dynamics of qubit 0. In Fig. 24 simulations with different maximal bond dimension are compared with the exact numerics for 10 qubits with such parameters and topology, leading us to set $\eta = 100$ for the rest of the simulations in this subsection, allowing a very high accuracy (errors can be estimated to be of order a percent in all quantities that we have access to). We also repeated a similar comparison (not shown) for a larger system ($N = 22$) and up to a long time ($t_f = 30$), with $\eta = 100$ and $\eta = 150$, finding a similar agreement.

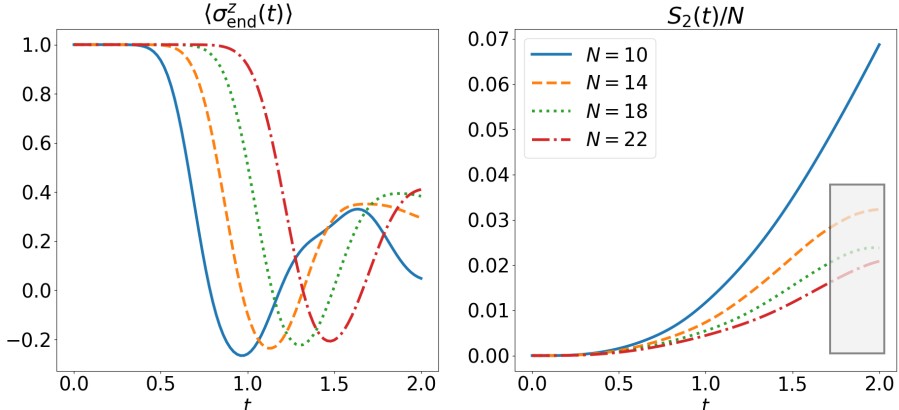

Figure 23: The $Z$ expectation value (left) and the second Rényi entropy $S_2$ per qubit (right) vs. the time $t$, of the end qubit at the right edge, for plaquettes of identical qubits as in Fig. 22, with the length varying between $N = 10$ to $N = 22$, and the parameters given in Eqs. (26)-(32). The relatively long-duration "dip" in $\langle Z \rangle$ observed with the end qubits of linear chains of different lengths (e.g., in Fig. 4), is not seen with this topology. The simulation for $N = 10$ is numerically exact, and $S_2$ can be seen to increase monotonously in this setup in the initial times, while for the MPO solutions with $N > 10$ and maximum bond dimension $\eta = 1000$, the $S_2$ curve manifests an inflection curve in its time dependence at $t \approx 1.7$. Marked with shading is the region where these simulations become relatively inaccurate – see the text regarding indications of simulation breaking.

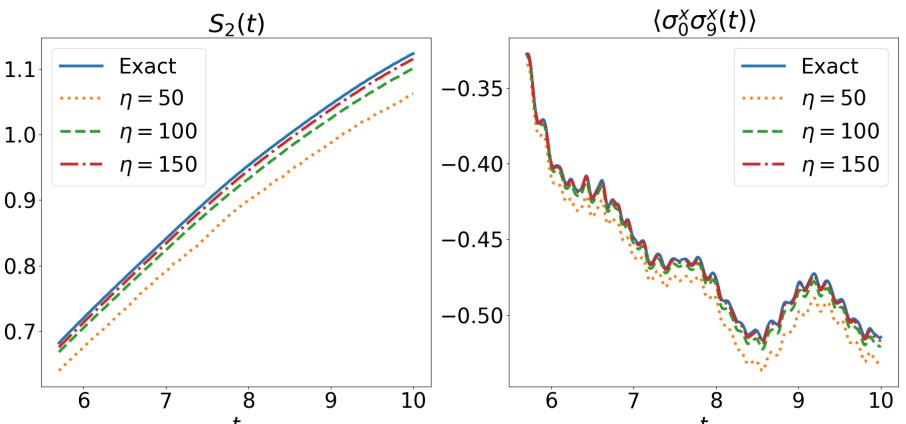

Figure 24: The second Rényi entropy $S_2$ (left), and the two-qubit $XX$ expectation value of the two edge qubits 0 and 9 (right), vs. the time $t$ (the earlier simulation times are not shown for clarity). A few solutions are presented for one plaquette of 10 alternating-frequency qubits as in Fig. 6, simulated by using both a numerically exact simulation (storing the full density matrix), and the MPO approximated simulation with different values for the maximal bond dimension $\eta$. Qubit 0 is the driven qubit, and the parameters are those of Eq. (33). From these simulations and similar ones (not shown) we conclude that $\eta = 100$ is sufficient for errors of order 1% in all studied quantities, and no qualitative problems in the dynamics.

# G The lab frame and rotating frame

In this section we derive the effective time-independent Hamiltonians by transforming from the lab frame to a rotating frame, and effecting some useful approximations. The derivation uses a notation of harmonic oscillator operators of creation ($a^\dagger$), annihilation ($a$) and number ($n$). The relations to the Pauli matrices for the restricted two-level system operators used in the text is straightforward;

$$n \to \frac{1}{2}(1 - \sigma^z), \qquad a^\dagger \to \sigma^-, \qquad a \to \sigma^+. \tag{43}$$

We start from the two-qubit Hamiltonian, with one qubit being driven in the lab frame,

$$H_0 = \omega_1 n_1 + \omega_2 n_2 + J(a_1^\dagger a_2 + a_1 a_2^\dagger) + \Omega_1 \cos(\nu_1 t + \phi_1)(a_1^\dagger + a_1). \tag{44}$$

Going to the rotating frame defined by the unitary

$$R = \exp\{-i\nu_1 n_1 t - i\nu_2 n_2 t\}, \tag{45}$$

where $\nu_2$ is to be specified later, the Hamiltonian in the new frame is

$$H = R^\dagger H_0 R - iR^\dagger \dot{R}, \tag{46}$$

with the second term being

$$-iR^\dagger \dot{R} = -(\nu_1 n_1 + \nu_2 n_2). \tag{47}$$

Using

$$a^\dagger n = (n-1)a^\dagger, \qquad a^\dagger n^2 = (n-1)^2 a^\dagger, \qquad \dots, \tag{48}$$

we get

$$a^\dagger e^{-i\nu nt} = e^{i\nu t} e^{-i\nu nt} a^\dagger, \qquad a e^{-i\nu nt} = e^{-i\nu t} e^{-i\nu nt} a. \tag{49}$$

Making the rotating wave approximation we get

$$R^\dagger H_0 R = \omega_1 n_1 + \omega_2 n_2 + \frac{1}{2}\Omega_1(e^{-i\phi_1} a_1^\dagger + \text{H.c.}) + J(e^{i\nu_1 t} a_1^\dagger e^{-i\nu_2 t} a_2 + e^{-i\nu_1 t} a_1 e^{i\nu_2 t} a_2^\dagger). \tag{50}$$

Defining

$$\Delta_1 = \omega_1 - \nu_1, \qquad \Delta_2 = \omega_2 - \nu_2, \qquad \nu_{12} = \nu_1 - \nu_2, \tag{51}$$

we have

$$H = \Delta_1 n_1 + \Delta_2 n_2 + J(e^{i\nu_{12} t} a_1^\dagger a_2 + \text{H.c.}) + \frac{1}{2}\Omega_1(e^{-i\phi_1} a_1^\dagger + \text{H.c.}). \tag{52}$$

In the case of a single qubit driven on resonance, we can set $\nu_1 = \nu_2 = \omega_1$ and we get the time independent Hamiltonian

$$H = \Delta_2 n_2 + J(a_1^\dagger a_2 + \text{H.c.}) + \frac{1}{2}\Omega_1(e^{-i\phi_1} a_1^\dagger + \text{H.c.}), \tag{53}$$

with $\Delta_2 = \omega_2 - \omega_1$, and qubit 1 has zero frequency. The above expression can be easily generalized in the case of multiple driven qubits, as long as there is just one drive frequency (here, $\omega_1$). Otherwise the resulting Hamiltonian remains time-dependent.

# H   Idle qubits with a large frequency difference

For the case of two idle (non-driven) qubits, we can set $\nu_1 = \omega_1$ and $\nu_2 = \omega_2$ (going to the frame where each qubit is rotating at its natural frequency), and then

$$\nu_{12} = \omega_1 - \omega_2, \tag{54}$$

leaving only the interaction Hamiltonian

$$H_I = J(e^{i\nu_{12}t}a_1^\dagger a_2 + \text{H.c.}). \tag{55}$$

A straightforward approximation of the dynamics exists in the limit of a large difference between the qubit frequencies. We use the expansion of the effective Floquet Hamiltonian from [70], justified for $J \ll |\nu_{12}|$, to obtain

$$H_{\text{eff}} \equiv \frac{1}{\nu_{12}}[H_1 H_{-1} - H_{-1}H_1], \tag{56}$$

with

$$H_m = \frac{1}{T}\int_0^T dt\, e^{-im\nu_{12}t}H_I, \qquad T = 2\pi/\nu_{12}, \tag{57}$$

and explicitly we have

$$H_1 = a_1^\dagger a_2, \qquad H_{-1} = a_2^\dagger a_1, \tag{58}$$

which gives

$$H_{\text{eff}} = \frac{J^2}{\nu_{12}}\left(a_1^\dagger a_2 a_2^\dagger a_1 - a_2^\dagger a_1 a_1^\dagger a_2\right), \tag{59}$$

or,

$$H_{\text{eff}} = \frac{J^2}{\nu_{12}}[n_1(n_2+1) - n_2(n_1+1)] = \frac{J^2}{\nu_{12}}[n_1 - n_2], \tag{60}$$

i.e., only a small frequency correction remains. A more interesting case is that of three qubit in a chain, with the frequencies of the first and third qubit being equal. Generalizing the transformation above to three qubits we have

$$H_I = J(e^{i\nu_{12}t}a_1^\dagger a_2 + e^{i\nu_{23}t}a_2^\dagger a_3 + \text{H.c.}), \tag{61}$$

with

$$\nu_{12} = \omega_1 - \omega_2 = -\nu_{23}. \tag{62}$$

We can therefore take $\nu_{12}$ as the fundamental frequency and write

$$H_1 = a_1^\dagger a_2 + a_3^\dagger a_2, \qquad H_{-1} = a_2^\dagger a_1 + a_2^\dagger a_3, \tag{63}$$

and by expanding $H_{\text{eff}}$ of Eq. (56), and collecting the terms that do not cancel, we get both the frequency shift of each qubit (according to the number of bonds it has), and an induced XY coupling between the pairs of next-nearest neighbor qubits,

$$H_{\text{eff}} = \frac{J^2}{\nu_{12}}\left[H_z + H_{xy}\right], \tag{64}$$

with

$$H_z = n_1 - 2n_2 + n_3, \qquad H_{xy} = a_1^\dagger a_3 + a_3^\dagger a_1. \tag{65}$$

Generalizing the derivation above to four qubits and more, one can easily see that no new terms result (beyond those that have identical nature on the additional bonds). The reason is that all terms with fours different qubits in the product $H_1 H_{-1} - H_{-1}H_1$ will cancel out since the four qubit operators all commute, and appear with inverse signs in the expansion.

# I Python parameters for the solver

In this section we detail the parameters supported for input by the solver's Python interface (at the time of preparation of this manuscript). For the full and up-to-date documentation of the solver input, output, and call interface, we refer the reader to the solver repository itself [43]. However, since the parameters are extensively discussed in this manuscript, we think it would be useful to explicitly list those parameters here. If there is a default value, it is given with an equality sign, otherwise it must be specified and an exception is thrown if it is not passed.

**Basic parameters**

1. `N`. The number of qubits in the lattice. This solver requires $N > 2$.

2. `t_final`. The final simulation time, $t_f$.

3. `tau`. The discrete time step $\tau$ used in the time evolution.

4. `t_init = 0`. The initial simulation time, $t_0$. Must obey $t_0 \leq t_f$.

5. `output_files_prefix = 'lindblad'`. The path and file name prefix to be used for the input file generated for the solver, as well as output files generated by the solver. Files will be appended with a unique id string (only if the parameter `b_unique_id` is set to `True`, see below), an indication of the number of qubits in the form `.N=8`, and suffixes indicating file types. The input file suffix is `.input.txt` and other types are indicated below.

6. `b_unique_id = False`. If `True`, a unique id will be generated for the simulation and appended to all generated file name prefixes. The unique id will be included in the parameters passed to the solver.

**Hamiltonian coefficients**

1. `h_x = 0`. The $h_{x,i}$ coefficient in Eq. (3). If a vector is given, it specifies $h_{x,i}$ for each qubit. If a scalar is given, it is uniform for all qubits.

2. `h_y = 0`. The $h_{y,i}$ coefficient in Eq. (3). The syntax and usage are identical to that of `h_x`.

3. `h_z = 0`. The $h_{z,i}$ coefficient in Eq. (3). The syntax and usage are identical to that of `h_x`.

4. `J_z = 0`. The $J_{ij}^z$ coefficient in Eq. (4). If a matrix is given, it specifies $J_{ij}^z$ for each pair of qubits. If a scalar is given, it is uniform for all qubits of a lattice, as specified below.

5. `J = 0`. The $J_{ij}$ coefficient in Eq. (4). The syntax and usage are identical to that of `J_z`. If either one of `J` or `J_z` is a matrix, then the other one must be either a matrix as well, or 0.

**Dissipation coefficients**

1. `g_0 = 0`. The $g_{0,i}$ coefficient in Eq. (6). The syntax and usage are identical to that of `h_x`.

2. `g_1 = 0`. The $g_{1,i}$ coefficient in Eq. (7).

3. `g_2 = 0`. The $g_{2,i}$ coefficient in Eq. (8).

**Initial state**

1. `init_pauli_state = '+z'`. Either a length-$N$ vector of two-character strings of the form $\pm a$, or a single such string. Each string indicates the initial state of qubit $i$ is a Pauli state as detailed in Sec. 2.2, and a single string indicates an identical initial state for all qubits.

2. `init_graph_state = []`. A list of integer tuples that specify the qubit pairs for performing a controlled-$Z$ gate on, to generate an initial graph state (starting with all qubits pointing along the $+x$ axis), as in Eq. (11).

3. `load_files_prefix = ''`. The prefix of files as previously saved using the simulator, which the initial state has to be loaded from. An empty string indicates that the initial state is not loaded. See the parameter `b_save_final_state` for more details on the saved files. If this string is nonempty, then `init_pauli_state` and `init_graph_state` must be an empty strings.

**Lattice specification**

If both parameters `J` and `J_z` specifying the qubit couplings are scalar (and not a matrix), then the simulator generates a uniform lattice coupling in embedded in $xy$ plane. The supported configurations are a 1D chain or a 2D rectangular strip (both with open boundary conditions), a 1D ring (with periodic boundary conditions), or a 2D cylinder (with periodic boundary conditions along the $y$ direction), specified using the following parameters.

1. `l_x = 0`. The length of the lattice along the x dimension ($l_x$). If left at the default 0 value, then the number qubits `N` is used, and `l_y` must take its default value (1) as well.

2. `l_y = 1`. The length of the lattice along the y dimension ($l_y$).

3. `b_periodic_x = False`. Whether periodic boundary conditions are applied along the $x$ dimension. If `True`, then `l_y = 1` is required.

4. `b_periodic_y = False`. If `True`, periodic boundary conditions in the $y$ direction are used.

**Numerical simulation control**

1. `trotter_order = 4`. Trotter approximation order, Possible values are 2, 3, 4.

2. `max_dim_rho = 400`. Maximum bond dimension for density matrices.

3. `cut_off_rho = 1e-16`. Maximum truncation error (discarded Schmidt weight) for density matrices. The actual truncation is done using the most severe condition between `cut_off_rho` and `max_dim_rho`.

4. `b_force_rho_trace = True`. Whether to force the density matrix trace to one by substituting $\rho \to \rho/\text{tr}\{\rho\}$ at every time step, compensating for finite-step errors.

5. `force_rho_hermitian_step = 4`. Determines every how many evolution time steps ($\tau$), to substitute $\rho \to (\rho + \rho^{\dagger})/2$. This may reduce some errors, but is computationally expensive.

6. `b_initial_rho_compression = True`. Whether a density matrix that is loaded from a previously saved state, should be re-gauged and compressed using the `orthogonalize()` method of the underlying ITensor MPS.

**Observables and output**

1. `b_save_final_state = False`. Whether to save the final state to files. Three binary files will be saved, whose names will be constructed from the value of `output_files_prefix` using different extensions. These files can be loaded to set the initial state for a new simulation using the parameter `load_files_prefix`.

2. `output_step = 1`. How often (in discrete steps of time `tau`) the observables are computed. For 0 no observables will be computed.

3. `1q_indices = []`. A list of integers that specify the qubits for calculating single-qubit observables as in Eq. (12). If left empty it will default to all qubits.

4. `1q_components = ['Z']`. A list of strings that specify the Pauli observables to compute for all qubits given in `1q_indices`, and save using a file name ending with `.obs-1q.dat`.

5. `2q_indices = []`. A list of integer tuples that specify the qubit pairs for calculating single-qubit observables as in Eq. (13). If left empty it will default to all qubit pairs.

6. `2q_components = ['ZZ']`. A list of strings that specify the two-qubit Pauli observables ('XX', 'XY', etc.), to compute for all qubits given in `2q_indices`, and save using a file name ending with `.obs-2q.dat`.

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
