# Peer review of "Nonlocal correlations in noisy multiqubit systems simulated using matrix product operators"

_SciPost Physics Core, doi:SciPost Phys. Core 6, 037 (2023)_

## Round 1 · Referee Report · Anonymous (Referee 1) · 2022-4-8

Strengths
1- I found the paper well written. 2- The open source library is properly documented on the git hub repository. 3- Solving the dynamics of a driven-dissipative quantum many-body system is a central problem in condensed matter and AMO physics. 4- I consider the idea of making open source, well documented, user-friendly codes for this class of systems a very important step in the study of many-body open system dynamics.
Weaknesses
1- I am not completely satisfied about the simulation part of the manuscript, Sec.3. See Report for the details.
Report
I found the paper well written and the open source library properly documented on the git hub repository. Solving the dynamics of a driven-dissipative quantum many-body system is a central problem in condensed matter and AMO physics. Many methods have been proposed in the last years and tensor-network methods have been proved to be very effective on a wide variety of physical problem. I thus consider the idea of making open source, well documented, user friendly codes for this class of systems a very important step in the study of many-body open system dynamics.
I do not have any particular remark about the structure true of the paper that I found very coherent and pedagogical. However I am not completely satisfied about the simulation part of the manuscript (Sec.3). In particular, I see that all the problem they choose to solve have an XY model as unitary part of the nearest-neighbour dynamics. After Jordan-Wigner transformations this dynamics maps onte free-fermion Hamiltonian in 1D.
In conclusion the paper could meet the publication Criteria of SciPost Physics after some revision on the simulation part I detailed below.
Requested changes
1- I propose to the authors to study also the effect of adding the interaction given by an ZZ coupling. This would make the problem “Interacting” and interesting effects may be seen in the deformation of the light come as well as in the entanglement properties (Rényi entropy and OSEE). 2-Furthermore, since the authors propose a “new” solver, I suggest to compare the results of their simulation with some numerically exactly-solvable case. The most simplest case that I do have in mind is the 1D Ising model with decoherent spin flip. See Fig.2 of Vicentini et al., Phys. Rev. Lett. 122, 250503 (2019).
We would like to thank the referee for reading the manuscript and for her/his positive feedback and suggestions.
Regarding the referee's first suggestion - to add ZZ coupling to our numerical studies (mostly focused on XY coupling), we have followed the suggestion and added simulations of this type to the manuscript. We also note that the referee's comment on the free-fermion dynamics is exact at the bulk of the system, but does not strictly hold for the first qubit (being driven), and in the plaquette case the integrability supposedly breaks also at the two lattice points where a qubit interacts with three neighbors. The overall system dynamics with the drive and noise are complex and interesting.
Following the referee's suggestion we have considered the effect of small ZZ coupling terms on the dynamics of nonlocal correlations that form an important result of the paper. Perturbative ZZ interactions are a significant source of uncontrolled dynamics in superconduting devices and the coupling strengths we chose for the simulation are realistic in magnitude. The results are discussed in Sec. 3.4 (towards the end) and in Sec. 4, and presented in detail in App. G (Figs. 22-23). Indeed the extra interactions make the dynamics more complex as is evident in the entanglement properties, and when the ZZ interaction is large enough the nonlocal correlations are interestingly washed away.
Regarding the referee's second suggestion, we indeed implemented it and added in Appendix C a study of the quantum Ising model on a ring with strong dissipation. The results coincide with those of the reference pointed out by the referee. We note also that, as detailed in the manuscript, we have systematically compared the solver results with independent numerical simulations for all setups studied in the paper (for up to 10 qubits), which realizes a comprehensive numerical test of the solver.

Author: Haggai Landa on 2022-08-12 [id 2724]
(in reply to Report 2 on 2022-05-31)We would like to thank the referee for pointing out the fact that the results are solid and that the paper is well written, and for his/her suggestions.
Regarding the first weakness pointed out by the referee (physical motivation for the work), we agree that this point was not properly addressed in the manuscript previously, and following the referee's comment we have now clarified the motivation and the relevance of the results to current state-of-the-art quantum hardware. As now emphasized in the introduction and conclusion sections, the models studied in the work consist a first step towards simulating many body dynamics of quantum devices. In essence, resonant driving of single qubits is the fundamental workhorse of quantum information processing in many qubit devices, being employed to realize single-qubit rotations and generate entanglement. The driven dynamics are often studied by focusing on small qubit systems, however, the focus of the current work is on the many body and nonlocal effects. Since uncontrolled nonlocal correlations of qubits in large devices could be detrimental to their usage for computational tasks, the ability to simulate and explore such aspects is important.
Regarding the second weakness (main result presented at the end), we have further emphasized this result in the introduction. Since the manuscript progresses towards this result by gradually increasing the complexity of the analysis and adding elements into the studied problem, it seems natural to us that the main result be at the end of the paper, which is an accepted choice in many publications.
Regarding the referee's comment that the main goal of the study seems to be benchmarking the code and that the paper should be published in SciPost Physics Codebases, we have a different opinion. The paper does not seem suitable for SciPost Physics Codebases since according to its website, a submission to Codebases consists of "a detailed userguide and a source code". Being neither of those, our paper is a proper research paper focused on two physical setups and new results presented together with a thorough analysis. With the numerical tool used for the study being new and state-of-the-art, the paper is meant to link numerical practices with the physical properties of the system and contribute to the readers' understanding of both aspects. Some important effort has been made to make the presentation pedagogical and it should be stressed that both referees indicated that the paper is indeed well written.
To conclude our reply, as supported by Report 1, we believe that the novelty of the results that we find and the relevance of the techniques to a multidisciplinary audience of researchers interested in open quantum system dynamics merit publication in SciPost Physics.

---

## Round 2 · Referee Report · Anonymous (Referee 1) · 2022-8-17

Report

The authors implemented the suggestions I pointed out in my first report. In my opinion the manuscript is now suitable for the publication in SciPost Physics in its present form. In particular the manuscript satisfies the criteria 3 : Open a new pathway in an existing or a new research direction, with clear potential for multipronged follow-up work;

---

## Round 2 · Referee Report · Anonymous (Referee 3) · 2022-9-23

Strengths

1- Link to a well written open source code 2- Some appendices are quite pedagogical and generally the paper is well written

Weaknesses

1- No new surprising discovery 2-Structure is a bit confusing

Report

Unfortunately I don't see any of the acceptance criteria for SciPost Physics fulfilled.

While it is nice that the paper introduces an accessible open code, I agree with the comment of a previous report: The paper demonstrates the applicability of the code for computing dynamics in two toy-model setups, but it does not present/analyze any real surprising physics discovery. I find this also to be true after the revision.

The discussion on the long-distance correlation (which could be an interesting lead) is too short and superficial, and it seems a bit lost after the "warm-up" simulations. Interesting questions would have been for example to analyze this correlation more deeply. Is there entanglement between the distant qubits? Probably this is a classical correlation given the large entropy and the plateau of the OSEE. I find an emergence of such a correlation for a specific mesoscopic setup not very surprising. Generally, I also think that the paper does not put these findings clearly into the context of the many known results of the vast literature on e.g. spreading of correlations.

Furthermore the connection to the IBM device is only vague ("a first step" towards a simulation as the authors describe it themselves). For an analysis of "unexpected correlations" in a real device, finite temperature would need to be included or the validity of the Lindblad approach would need to be tested.

Finally, also in terms of the numerical method there are no new algorithmic advances.

In conclusion, I don't think the paper has enough novelty for SciPost Physics and I don't see any potential for it to open up a new research direction. Nevertheless it is solid research and deserves a publication e.g. in SciPost Physics Core.

Requested changes

1- I'm a bit confused about the presentation of the Lindblad terms: rates g_0, g_1, and g_2 are introduced, but I think only g_0 and g_2 are used in the main text? Furthermore, when talking about "energy relaxation", I suppose they refer to spontaneous emission. The latter would be connected to g_1, so did the authors confuse g_0 and g_1, or is it just an unconventional definition of the spin-lowering/raising operators? This should be written more clearly.

2- Some text pieces are too vague, in particular:

Section 2.5: The part with the "1D path" geometry is quite hard to understand for someone not familiar with the method. A little sketch would help? Furthermore, here it would be more useful to say a sentence on the type of density-matrix linearization used, e.g. referring to Eq. (40).

Section 3.4, bottom paragraph on page 15: The descriptions here (about the OSEE) are very vague and hand-wavy. E.g. it's not clear how the authors distinguish classical and quantum correlations. Also the connection to the bond dimension does not become clear and is not entirely precise. I suggest to be either more mathematical or to cut pieces and replace them with references.

4- I suggest to modify the structure of the paper. There are a lot of appendices, some with interesting results on correlations (which I find more interesting than the chain results in Sec. 3.2). I would propose to make one clear section with the correlation results for the IBM-type setup, and one for other "benchmark-type" simulations, both in the main text.

  • validity: high
  • significance: low
  • originality: low
  • clarity: good
  • formatting: good
  • grammar: excellent

Author:  Haggai Landa  on 2022-11-25  [id 3071]

(in reply to Report 2 on 2022-09-23)

We would like to thank the referee for her/his reading of the manuscript and valuable feedback.

Following the referee's suggestion to quantify entanglement in the system, we have looked at the concurrence as a measure of nonseparability of the edge qubits and added a plot of its dynamics for different system sizes, together with a discussion of this point (in Sec. 4.2). The suggestion raised by the referee that the correlation is classical is correct in the steady state. This is an expression of the fragility of the entanglement, which is initially mediated by the propagating excitations that manifest rich dynamics. The eventual state, though not entangled, is nontrivial and correlated, and we think that this setup merits further exploration. Our results exemplify a state-of-the-art tool allowing to study the dynamics of correlation spreading in systems relevant in various subfields of open quantum systems.

For the referee's comment that "finite temperature would need to be included or the validity of the Lindblad approach would need to be tested", we note that energy exchange with a finite temperature Markovian bath is already supported in the model (since the Lindblad equation includes relaxation and excitation operators). In qubit devices the environment temperature is typically significantly lower than the qubit's energy gap, and hence keeping the spontaneous emission terms only (and dephasing of course) is typically an accurate approximation. We have added a comment regarding this point together with a reference.

For the referee's requested changes: 1. Indeed with superconducting qubits the Hamlitonian is often taken with a negative sign for the single-qubit \sigma^z (energy terms), as it appears in Sec. 3.1 of the paper, making |up> the qubit's ground state, and hence the spontaneous emission rate is g_0. We have clarified this source of confusion in the text. 2. About Section 2.5: we have improved the explanation of the "1D path". We have in particular made connection with Eqs. 41-42 (product of matrices) in App. A, and we are now also referring to a graphical illustration presented in Ref. 35. 3. About Section 4.1: The OSEE indeed does not tell if the correlations are mostly classical or quantum. In other words, it is not an entanglement measure. We added this remark as a footnote in Sec. 4.1. Regarding the lack of details concerning the connection between the OSEE and the bond dimension, this connection is in fact not entirely direct and making a comprehensive mathematical statement is not easy, going beyond the scope of the current discussion. We have added a reference that discusses this question (in the closely related framework of MPS). 4. We have implemented the referee's suggestion and created separate sections for the chain dynamics and for the plaquette dynamics. We have shortened the former and have incorporated some correlation results from the appendix into the latter.

To conclude, we wish to thank the referee again, and resubmit the amended manuscript to SciPost Physics Core.

---

## Round 2 · Author Response

Dear editor,
We would like to thank the referees for reading the manuscript and providing useful feedback.
Following the referees' suggestions we have improved the clarity of the manuscript and the analysis of the results, and added new numerical simulations that solidify our results.
We believe that the novelty of the obtained results and the importance of the techniques to a multidisciplinary audience in the field of open quantum system dynamics merit publication in SciPost Physics.
We provide a list of changes and we reply to each referee's report separately.
Kind regards,
the authors.

---

## Round 2 · List of Changes

• We have emphasized in the introduction and conclusion sections the relevance of the models studied in the work as being a first step towards simulating many body dynamics of quantum devices. We have clarified the discussion of the physical motivation and the main results in these sections.
  • We have added simulations of ZZ interaction (together with the XY coupling) to the manuscript. Perturbative ZZ interactions are a significant source of uncontrolled dynamics in superconduting devices we chose realistic coupling strengths accordingly. Thosee results are discussed in Sec. 3.4 (towards the end) and in Sec. 4, and presented in detail in App. G.
  • We added in Appendix C a study of the quantum Ising model on a ring with strong dissipation. The results coincide with those of earlier literature on this setup.

---

## Round 3 · Referee Report · Anonymous · 2023-1-10

Report

After the last modifications the manuscript has improved very much. It presents a nice solid research work that definitely fulfills the acceptance criteria of SciPost Physics Core. I think it can be published in its current form.

---

## Round 3 · Author Response

Dear editor,
We would like to thank again the referees of the previous refereeing round for reading our manuscript and writing their reports.
We have clarified the points pointed to by the second referee, added some new explanations and simulations, and restructured slightly the manuscript following his/her suggestion.
Despite the first referee's recommendation to publish the paper in SciPost Physics, we are following the second referee's recommendation to resubmit to SciPost Physics Core.
With kind regards,
The authors.

---

## Round 3 · List of Changes

In Section 4.2 we have looked at the concurrence as a measure of nonseparability of the edge qubits and added a plot of its dynamics for different system sizes, together with a discussion of this point.
In Section 2.5 we have improved the explanation of the "1D path".
We have clarified a few explanations and emphasized a few points along the paper, adding some references where needed.
We have created separate sections for the chain dynamics (Sec. 3) and for the plaquette dynamics (Sec. 4). We have shortened the former and have incorporated some correlation results from the appendix into the latter.

---

## Editorial Decision

published